# Environmental fluctuations alter the competitive trade-offs of group size in a social primate

Odd T. Jacobson [1,2] ✉, Margaret C. Crofoot [1,2,3], Genevieve E. Finerty [1,2,3], Susan E. Perry [4,5] & Brendan J. Barrett [1,2,3,5]

Larger animal groups are widely understood to require more space and expend more energy to mitigate the foraging costs of within-group competition. Yet between-group interactions and shifting resource distributions can obscure links between group size and behaviour, making responses to demographic change difficult to predict. Here, using 33 years of observational data from 12 neighbouring white-faced capuchin (*Cebus imitator*) groups in Costa Rica, combined with remotely sensed environmental data, we show that within- and between-group competition jointly shape space use, with their relative importance shifting with seasonal and interannual climate cycles. Larger groups compensated for reduced per capita foraging efficiency by expanding into less-exploited areas over longer timescales rather than increasing daily travel. Notably, this expansion disproportionately encroached on smaller neighbouring groups. In the dry season, resource confinement to riparian zones increased intergroup encounters and reduced overlap, with larger groups occupying the highest-quality areas. Climatic extremes linked to El Niño and La Niña exacerbated within-group foraging costs for large groups, whereas intermediate anomalies relaxed these constraints and amplified the benefits of between-group competitive ability. Our findings show that environmental variation shifts the trade-offs of within- and between-group competition, shaping how group-living animals adjust to changing social and ecological conditions.

Functional traits of animals (for example, body size, life-history strategy, social structure) carry both costs and benefits that depend on specific ecological conditions[1,2]. Understanding how environmental change alters the balance of these trade-offs is key to explaining their wide variation across and within species (and populations), as well as to forecasting evolutionary trajectories[3,4]. Group size (an emergent demographic trait) represents one such trade-off for group-living animals, as it affects both within- and between-group resource competition, but often in opposing ways[5–7].

Within-group competition imposes a cost by reducing individual foraging efficiency, thereby setting an upper limit on group size (that is, the ecological constraints model of group size[8–10]), which primarily reflects scramble (indirect) rather than contest (direct) competition among group members[5]. Consequently, larger groups face higher energetic costs from increased resource requirements and travel distances, driven by greater collective metabolic demands and faster resource depletion[11,12]. They also often require larger home ranges (for example,

---

[1]Department for the Ecology of Animal Societies, Max Planck Institute of Animal Behavior, Konstanz, Germany. [2]Department of Biology, University of Konstanz, Konstanz, Germany. [3]Center for the Advanced Study of Collective Behavior, University of Konstanz, Konstanz, Germany. [4]Department of Anthropology, University of California-Los Angeles, Los Angeles, CA, USA. [5]These authors jointly supervised this work: Susan E. Perry, Brendan J. Barrett. ✉e-mail: ojacobson@ab.mpg.de

refs. 13–16; but see ref. 17)—similar to how individual space-use scales allometrically with body size[18].

While larger groups typically face greater costs from within-group competition, they often hold a competitive edge in between-group contests, leveraging their numerical advantage to exclude smaller neighbours from key resources[17,19–22]. This advantage is especially pronounced when resources, such as fruiting trees, are distributed in defensible patches that vary in quality, allowing larger groups to monopolize the most valuable patches[6]. Larger groups also benefit indirectly, as smaller groups may avoid high-risk areas—granting larger groups access to a wider array of food patches with reduced opposition[23,24]. With more (and better) foraging options available, larger groups can access richer and less-depleted patches, potentially mitigating the costs of within-group competition and contributing to higher survival and reproductive success[6,25–27].

Ultimately, group size reflects trade-offs between the costs (for example, within-group competition, disease transmission) and benefits (for example, between-group competitive advantage, reduced predation risk) of group living[28,29]. Still, group sizes often vary widely within populations and fluctuate temporally, suggesting that this balance is mediated by additional ecological and social factors[30–32]. Demographic turnover reshapes the distribution of group sizes across the population, altering both ecological constraints and the balance of power among neighbouring groups[22,33]. Simultaneously, local heterogeneity and shifting environmental conditions influence both the intensity of within-group competition and the cost-effectiveness of resource defence[34–36]. Smaller groups may benefit when resources are abundant or widely distributed, making them less practical to defend—thereby reducing the importance of intergroup dominance[19]. By contrast, during resource-scarce periods, larger groups may gain an advantage by using their numerical superiority to outcompete smaller groups for remaining food patches[27,37]. This advantage, however, could be offset if larger groups incur disproportionately high costs from within-group competition under the same conditions[38]. Thus, which group sizes are favoured is probably context dependent, shifting with demographic and environmental change[4,28,39].

The ecological constraints model is a widely accepted framework for understanding how within-group competition limits group size in social animals[8,9,11,40–43]. As with any broad framework, however, real-world ecological complexity introduces important nuance. The interdependence of within- and between-group competition, along with the dynamic nature of landscapes and competitive interactions, often obscures causal links between group size and behaviour. Thus, how costs of within-group competition interact with benefits of between-group competition at the population level—particularly as resource availability fluctuates—remains unresolved. Clarifying how this interaction unfolds over time is critical for understanding the evolution of sociality, the ecological conditions under which it is favoured and its trajectory in a rapidly changing world. Such insight requires long-term datasets from multiple, neighbouring social groups that encompass meaningful demographic and ecological variation—data that are logistically difficult to obtain. Most previous research has therefore examined within- and between-group competition in isolation (often cross-sectionally), lacking either the temporal depth or number of social groups needed to explore their interaction.

Drawing upon 33 years of data on 12 white-faced capuchin monkey groups from the Lomas Barbudal Monkey Project in Costa Rica (10° 29–32′ N, 85° 21–24′ W), we examine how group size and environmental variation structure population-level patterns of resource competition in a group-living animal[44]. Lomas Barbudal lies within one of the last remaining fragments of tropical dry forest[45]—a biome characterized by a highly seasonal climate with a distinct wet season (May–November) and a harsh dry season (December–April) (Fig. 1b; ref. 46). Seasonal shifts dramatically alter resource distribution and availability. Most of the forest is deciduous; during the dry season, water, shade and food

become concentrated in evergreen riparian zones (Supplementary Note 3). The region is also highly sensitive to interannual climatic fluctuations associated with the El Niño-Southern Oscillation (ENSO; El Niño and La Niña; Supplementary Note 3.2; ref. 47). In the tropical Americas, El Niño events typically produce abnormally hot, dry conditions, whereas La Niña events bring abnormally cool, wet conditions[48].

This system offers a rare opportunity to study how long-term demographic and environmental fluctuations shape competition and space-use dynamics, owing to both the exceptional breadth of the dataset and the pronounced environmental variation in the region. Capuchins in this population live in cohesive multi-male, multi-female groups of 5–40 individuals (mean = 18.8; ref. 44) that persist for years or decades, although occasionally split permanently into two independent units with distinct home ranges (9 events over the study period; Fig. 1c). Capuchins are primarily frugivorous (~50–55%), with invertebrates comprising most of the remaining percentage (based both on time spent feeding[49] and proportion of food consumed[50]). Groups are range resident for decades and their home ranges overlap extensively (Fig. 1a). Intergroup contests are hostile and occur frequently (~1 per week per group)[51,52].

Here we integrate conventional group-level predictions of the ecological constraints model—linking group size to indicators of within-group competition—with dyadic comparisons of neighbouring groups to test how relative group size influences between-group relationships. First, we evaluate annual and seasonal relationships between group size and (1) fruit foraging efficiency, (2) daily path length, (3) return rate to previously used areas, (4) home range area and (5) home range quality. Next, we apply hierarchical social relations models (SRMs) to evaluate how relative group sizes within group-dyads predict home range overlap and encounter rates, and how these relationships vary with seasonal shifts in vegetation productivity (see Table 1 for biological interpretation of each response variable). Finally, we investigated how seasonal severity influences within- and between-group competition, with the aim of assessing how climatic anomalies linked to ENSO modify the competitive trade-offs of group size. Together, these analyses provide longitudinal tests of how group-living animals adjust both within- and between-group competition in response to group size and environmental change (see Supplementary Note 4 for a priori predictions and causal pathways). Our findings highlight the dynamic trade-offs of group living and show how ecological and social pressures jointly shape space-use strategies—factors that ultimately determine access to resources and fitness.

## Results

### Larger groups face foraging costs but do not increase daily travel

To evaluate how group size influences resource competition, we combined long-term movement and observational data with continuous-time movement modelling (ctmm) and Bayesian generalized linear multilevel models (GLMMs) to estimate five response variables: (1) per capita daily fruit intake rate, (2) daily path length, (3) mean revisitation rate across the home range, (4) home range area and (5) home range quality, assessed via normalized difference vegetation index (NDVI)—a satellite-derived measure of vegetation greenness. All GLMMs were run at both annual and seasonal scales.

We found strong evidence for a negative association between group size and per capita fruit intake rate, but little to no association with daily path length in either season (Fig. 2a). Revisitation rates were considerably lower in larger groups at the annual scale and during the wet season ($\beta_{annual} = -0.11$ [−0.23, 0.00]; proportion of the posterior (PP) >0 = 0.07; $\beta_{wet} = -0.22$ [−0.34, −0.09]; PP > 0 = 0.008), but showed no relationship in the dry season ($\beta_{dry} = -0.06$ [−0.2, 0.08]; PP > 0 = 0.262). Larger groups consistently occupied larger home ranges across both scales, with a stronger effect observed in the wet season ($\beta_{wet} = 0.36$ [0.24, 0.49]; PP > 0 = 1; $\beta_{dry} = 0.27$ [0.13, 0.41]; PP > 0 = 0.998).

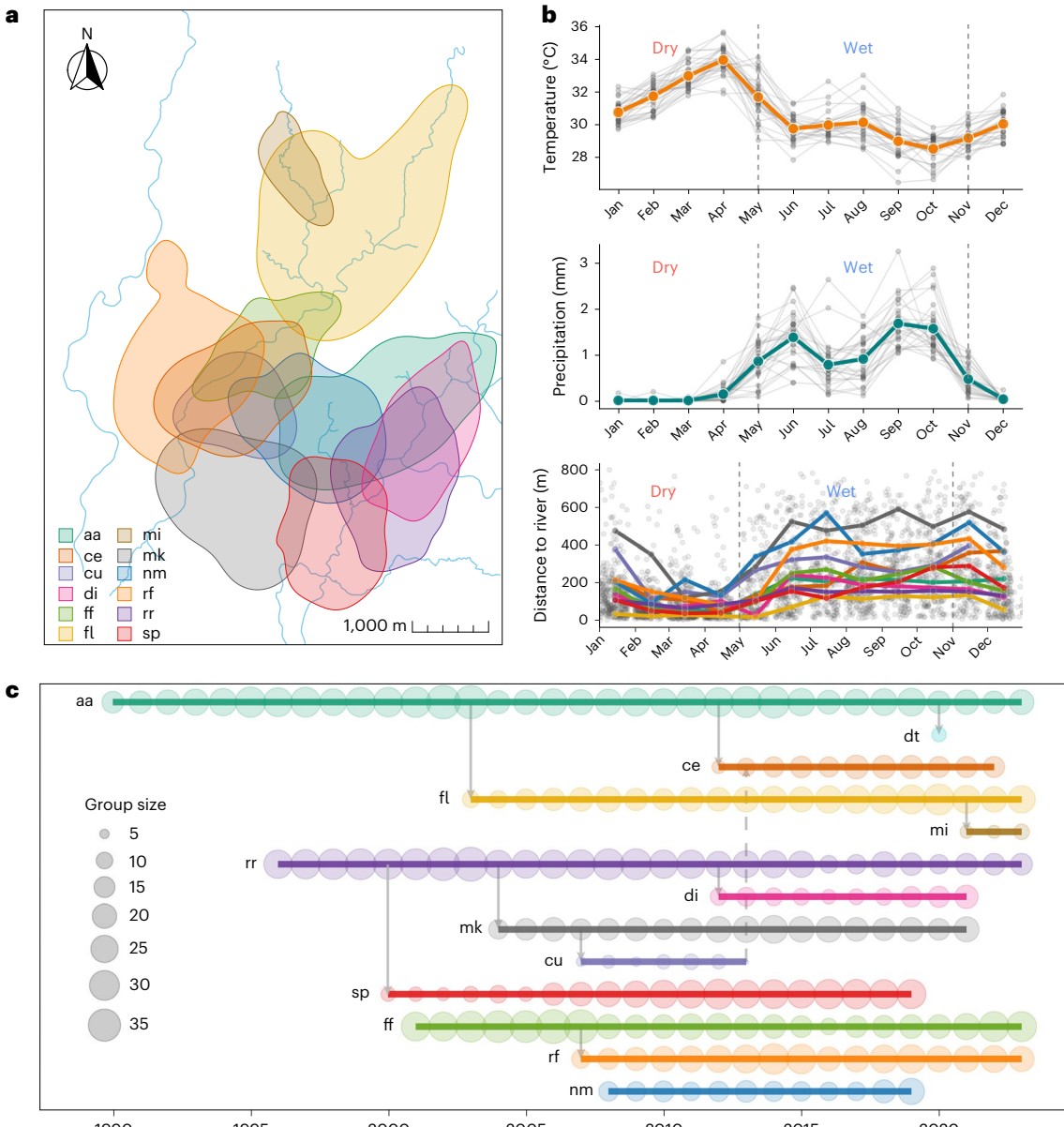

**Fig. 1 | Environmental and demographic context. a,** Annual home ranges of 12 white-faced capuchin study groups (95% auto-correlated kernel density contours) at the Lomas Barbudal Monkey Project, Costa Rica. Colours correspond to group identity; group names are indicated by the two-letter abbreviations shown in the legend. Estimates are derived from handheld GPS tracking data collected in 2012, except for ce and di (formed after fissions in late 2012; shown using 2013 data) and mi (formed after a 2021 fission; shown using 2021 data). Group dt is not shown owing to insufficient location data. **b,** Weather data near the Lomas Barbudal Biological Reserve from the ERA5 reanalysis dataset (1990–1993, 1997–1999, 2001–2020). The top panel shows temperature, the middle panel shows total precipitation and the bottom panel shows each study group's mean distance to the river. For temperature and precipitation, grey lines and points show monthly means per year; coloured lines and points (orange for temperature, blue-green for precipitation) show the monthly mean across years. For distance to river, grey points represent the mean daily distance to the river; coloured points and lines show monthly means by group. Dashed vertical lines mark typical transitions between dry and wet seasons. **c,** Time series of data collection for all 13 study groups from 1990 to 2023. Group abbreviations indicate monitoring start dates or formation via permanent fission. Arrows denote fission events (solid) and a single fusion event (dashed).

## Seasonal severity alters group-size effects on range quality and fruit intake

We quantified seasonal water-balance anomalies using the 6-month standardized precipitation–evapotranspiration index (SPEI-6). At our site, SPEI-6 is strongly correlated with the multivariate ENSO index (MEI; Supplementary Fig. 5), indicating that local hydroclimatic anomalies are largely ENSO linked. We therefore treat SPEI-6 as our primary measure of local climate anomalies, interpreting negative values as El Niño like (drier than average) and positive values as La Niña like (wetter than average).

For home range quality, larger groups generally occupied areas with higher NDVI in the dry season, but not in the wet season (Fig. 2a; $\beta_{dry} = 0.02$ [0, 0.05]; PP > 0 = 0.94; $\beta_{wet} = 0$ [−0.02, 0.02]; PP > 0 = 0.545). This positive relationship was strongest during 'exceptionally wet' dry seasons (La Niña like; SPEI = 1.5) and near zero during 'exceptionally dry' dry seasons (El Niño like; SPEI = −1.5; Fig. 2b). By contrast, during the wet season, the effect of group size on home range NDVI was inconsistent across the SPEI gradient.

For per capita fruit intake, group size had an overall negative effect across both seasons (Fig. 2a). This negative relationship was

## Table 1 | Summary of response variables

| Response variable | Abbreviation | Level | Competition type | Interpretation |
|---|---|---|---|---|
| Per capita fruit intake rate | FIR | Individual | Within group | Individual energy intake from fruit |
| Daily path length | DPL | Group | Within group | Daily energy expenditure |
| Revisitation rate | RR | Group | Within group | Frequency of returns to previously used areas |
| Home-range area | HRA | Group | Within group | Long-term space requirement |
| Home-range NDVI | HRQ | Group | Between group | Home-range quality |
| Proportional overlap | $PO_{fn}$ | Group-dyad (asymmetric) | Between group | Extent of neighbour access to focal range |
| Encounter rate | ER | Group-dyad (symmetric) | Between group | Frequency of intergroup encounters |

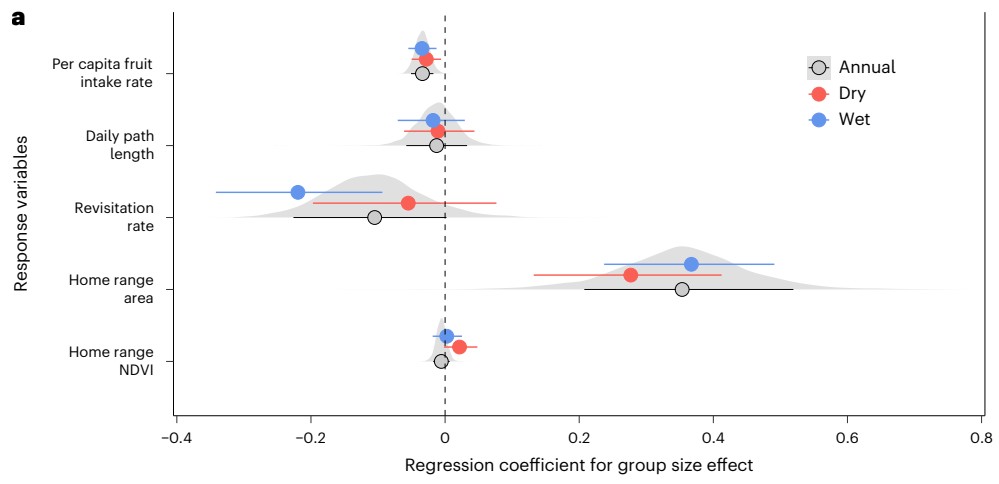

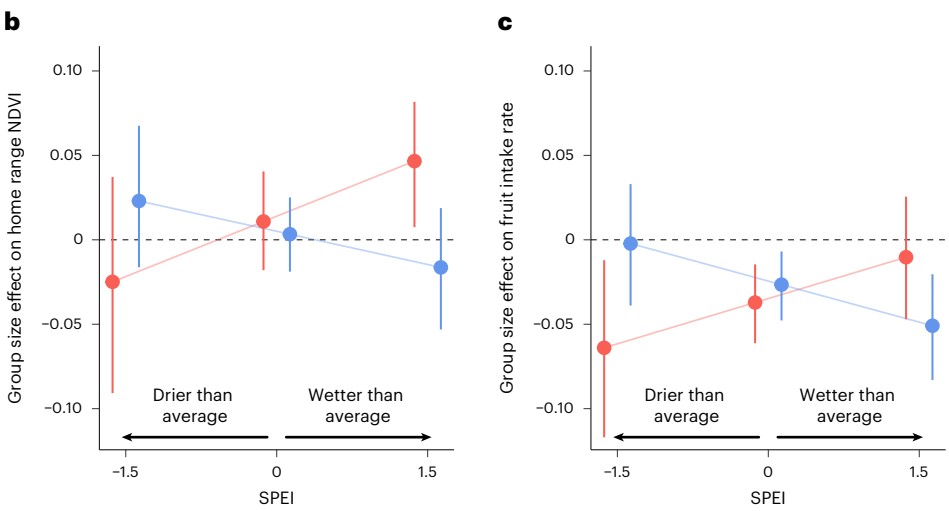

**Fig. 2 | Group size effects on resource and space use. a**, Posterior estimates of the marginal effect of group size on five group-level response variables: per capita fruit intake rate, daily path length, revisitation rate, home range area and home range NDVI. Points indicate median posterior estimates, and solid lines represent 89% HPDIs. Blue and red point intervals correspond to wet and dry season effects, respectively, while black and grey intervals indicate annual (non-seasonal) effects. Grey density slabs show the full posterior distributions for the annual effects. **b,c**, Point intervals show posterior estimates of the effect of group size on mean home range NDVI (**b**) and per capita fruit intake rate (**c**) conditional on representative SPEI values (−1.5, 0, 1.5), illustrating the three-way interaction between group size, season and seasonal severity (SPEI). Points are posterior medians with 89% HPDIs (vertical bars). Blue and red denote wet- and

dry-season estimates, respectively. Lines connect median effects across SPEI values to aid visual comparison. In both panels, *y*-axis values greater than 1 indicate a positive group-size effect (larger groups have higher NDVI and intake) and values less than 1 indicate a negative effect. On the *x* axis, negative SPEI values represent abnormally dry periods and positive values represent abnormally wet periods, relative to typical conditions within each season (January–April = dry; May–December = wet). Fruit intake models used *n* = 335 focal individuals across 11 groups; daily path length models used *n* = 996 group-days across 11 groups; home range area and NDVI models used *n* = 156 annual and *n* = 224 seasonal home ranges across 12 groups; revisitation rate models used *n* = 101 group-years annually and *n* = 143 group-season-years seasonally across 11 groups.

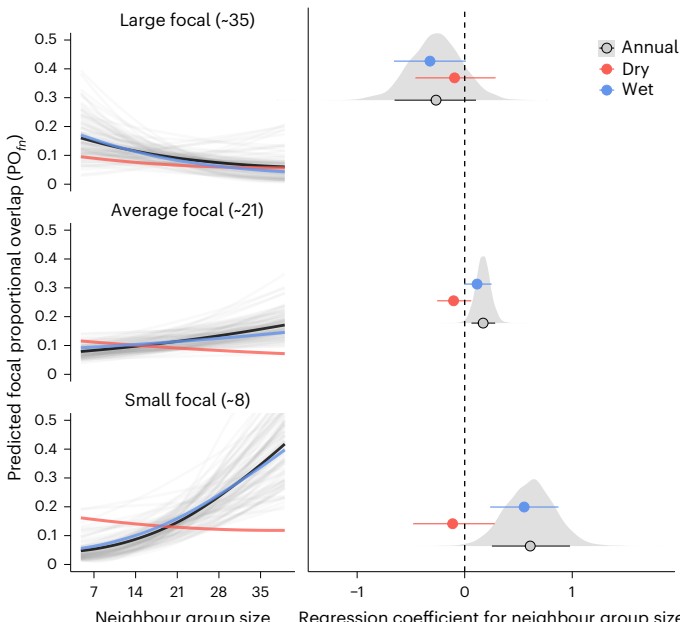

**Fig. 3 | Relative group size effects on neighbour encroachment.** The left panels show posterior predictions of proportional focal home range overlap ($PO_{fn}$) as a function of neighbouring group size across representative focal group sizes. Light grey lines depict 100 randomly sampled posterior draws from the annual model; dark black lines show the median annual prediction; red and blue lines show median predictions for the dry and wet seasons, respectively. The plot on the right shows median posterior effect sizes and 89% HPDIs for both annual and seasonal models, corresponding to the predictions in the left panels. Grey density slabs are posterior distributions for the annual effects. The annual model was fit to $n = 930$ dyad-years (59 dyads, 12 groups); the seasonal model used $n = 978$ dyad-season-years (56 dyads, 12 groups). Each dyad contributes two observations per year or season-year to account for outcome asymmetry.

most pronounced under climatically extreme conditions: during 'very dry' dry seasons (El Niño like; SPEI = −1.5) and 'very wet' wet seasons (La Niña like; SPEI = 1.5; Fig. 2c). When climatic anomalies counterbalanced the typical seasonal pattern—wet seasons that were unusually dry (El Niño like; SPEI = −1.5) or dry seasons that were unusually wet (La Niña like; SPEI = 1.5)—the effect of group size on fruit intake was near zero.

### Larger neighbours encroach onto smaller focal groups

To evaluate how group size asymmetries shape intergroup spatial dynamics, we modelled proportional home range overlap between neighbouring group-dyads using hierarchical SRMs[53,54] within a causal inference framework[55]. This approach accounts for repeated measures within and across dyads by simultaneously estimating group-level tendencies (for example, a group's general propensity to overlap with others) and dyad-specific effects (for example, the unique overlap between a particular pair). We incorporated group-level predictors (focal and neighbour group size) and a dyad-level predictor (interaction between focal and neighbour group size) to explain variation in asymmetric (that is, directional) overlap ($PO_{fn}$), defined as the proportional home range overlap of a focal group $f$ with a specific neighbour $n$. By adjusting for the home range area of $f$, $PO_{fn}$ can be interpreted as the degree of encroachment by the neighbouring group onto the focal group's range. While SRMs are typically applied to dyads of individuals, extending the framework to dyads of animal groups (1) is key to understanding how within-group and between-group competition jointly shape behaviour and (2) provides a new tool for assessing group-level social networks.

Using this approach, we found that an increase in neighbour group size usually led to an increase in $PO_{fn}$ (Fig. 3). However, this relationship was strongest when the focal group was small (~8 monkeys; $\beta_{PO} = 0.61$ [0.25, 0.98]; PP > 0 = 0.995), moderate when the focal

group was intermediate in size (~21 monkeys; $\beta_{PO} = 0.17$ [0.06, 0.28]; PP > 0 = 0.992) and weakly negative when the focal group was also large (~35 monkeys; $\beta_{PO} = -0.26$ [−0.65, 0.11]; PP > 0 = 0.128). This pattern held when examined annually and in the wet season, but not in the dry season, during which $PO_{fn}$ was generally reduced and neighbour group size had minimal impact (Fig. 3).

To better understand whether smaller or bigger groups in a dyad drove temporal increases in overlap (that is, change in $PO_{fn}$ at time $t$ to $t + i$), we conducted a complementary analysis of dyads that showed both substantial increases in annual $PO_{fn}$ (from <25% to >45%) and shifts in relative group size (net change ≥5) (see 'Dyad-level response variables' in Methods). In 84% of cases, the group that caused the overlap increase was the one that became relatively larger over time (Fig. 4), consistent with a competitive advantage of larger groups for space.

### Dry conditions increase encounters and reduce overlap

To investigate how seasonality affects intergroup encounter rates, we used the `ctmm` framework to estimate expected space–time overlap between group home ranges (that is, encounter rate)[56]. We then fit dyadic Bayesian multiple-membership models to assess seasonal variation in encounter rates as a function of symmetrical (that is, non-directional) dyadic predictors: absolute group size difference, overlap zone area and vegetation greenness within the overlap zone (measured by mean NDVI). Predicted encounter rates were consistently higher in the dry season than in the wet season, despite reduced dyadic

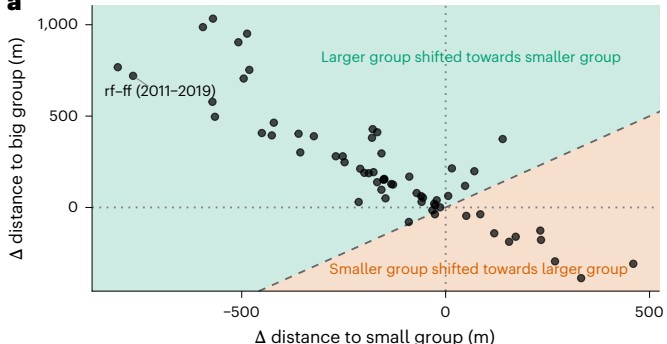

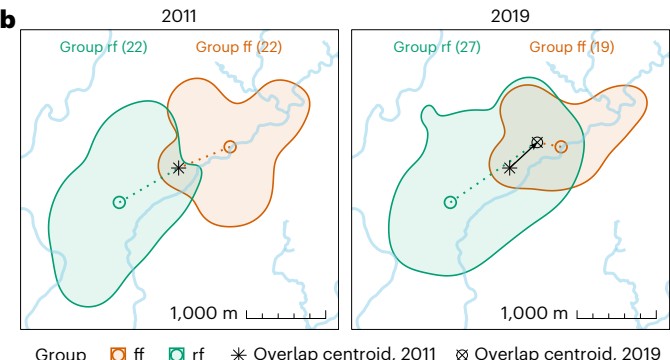

**Fig. 4 | Directional shifts in home range overlap. a,** Each point represents a dyad–year pair with a marked increase in overlap and a substantial shift in relative group size. Axes show the change in distance ΔD from each group's home range centroid at time $t$ to the overlap centroid at time $t + i$, relative to the original distance at $t$. The $x$ axis shows ΔD for the group that became relatively smaller, the $y$ axis for the group that became larger. Positive values indicate that the overlap centroid moved away; negative values indicate that it moved closer. Shaded colours indicate which group drove the shift. The dashed diagonal marks equal shifts. **b,** Example of dyad rf–ff (2011–2019) showing the overlap centroid's shift. Bold contour lines represent 95% home range boundaries; large circles show group centroids. Dotted lines connect group centroids to overlap centroids; the arrow shows directional change. Point shapes indicate overlap centroid locations; group sizes are in parentheses.

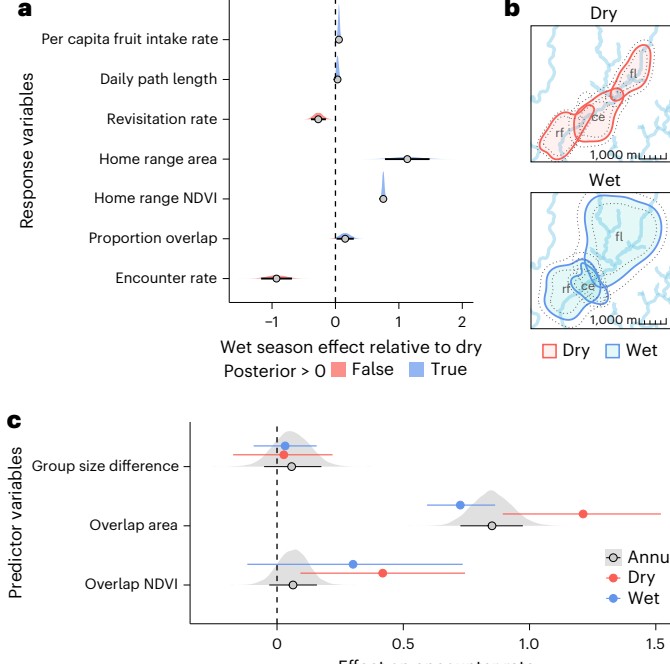

**Fig. 5 | Seasonal changes in fruit intake, ranging behaviour and encounter rates. a**, Posterior densities, medians and 89% HPDIs for wet versus dry season contrast across all seven response variables: fruit intake, path length, revisitation rate, home range size, home range NDVI, proportional overlap (PO_fn) and encounter rate. Red shading indicates the portion of the posterior density falling below zero (dry season effect); blue shading indicates the portion falling above zero (wet season effect). **b**, Seasonal comparison of home range overlap among three groups in 2018. Solid contour lines indicate mean 95% home range boundaries; dotted lines represent the lower and upper 95% confidence intervals. **c**, Posterior medians and 89% HPDIs for annual and seasonal effects of group size difference, overlap size and overlap NDVI on encounter rate. Blue and red point intervals correspond to wet and dry season effects, respectively, while black and grey intervals indicate annual (non-seasonal) effects. Grey density slabs are posterior distributions for the annual effects. Sample sizes for fruit intake, path length, revisitation rate and home range models are as reported in Fig. 2; overlap model sample sizes are as reported in Fig. 3. Encounter rate models were fit to $n = 269$ dyad–years (42 dyads, 11 groups) annually and $n = 280$ dyad–season–years (41 dyads, 11 groups) seasonally. Note that seasonal and annual models are fit to different subsets of the data, which may contribute to discrepancies between seasonal and annual effect sizes.

overlap (Fig. 5a). This pattern is consistent with a stronger effect of overlap zone size on encounter rate in the dry season, indicating more frequent encounters per unit of shared space compared with the wet season (posterior contrast: $\beta_{wet-dry} = -0.49$ [−0.78, −0.16]). Encounter probability was highest in overlap zones with greater vegetation productivity, particularly in the dry season ($\beta_{HRQ} = 0.42$ [0.09, 0.74]). By contrast, absolute group size difference had no consistent effect on encounter rate across both scales (Fig. 5c).

## Discussion

Our findings show that trade-offs of group living are shaped by complex interactions between environmental conditions and group size, which jointly mediate how within- and between-group competition influence the social and spatial structure of a population. Larger capuchin groups showed reduced per capita fruit intake efficiency, indicating stronger within-group competition over clumped resources—consistent with the ecological constraints model[8]. However, this increased competition did not translate into higher daily travel costs for larger groups (contra common expectation; for example, refs. 11,12). Instead, they appear to mitigate foraging competition by leveraging their numerical advantage

to outcompete smaller groups, gaining access to less frequently visited areas during the wet season (when food is abundant and dispersed) and to higher-quality patches during the dry season (when resources are concentrated in riparian zones).

The transition from wet to dry season may mark a shift from indirect to more direct between-group competition, as resources become limited and more economically defensible. Supporting this, we found that home range overlap declines in the dry season—consistent with increased active defence[57]—while intergroup encounter rates rise, particularly in areas with higher vegetation productivity near permanent water sources. This pattern matches theoretical models predicting that frequent encounters can drive spatial separation, as repeated costly interactions discourage intrusion into contested areas, thereby promoting more fixed home range boundaries and reducing resource loss to neighbours[58,59]. Reduced overlap and the tendency for larger groups to occupy more productive ranges in the dry season suggest that increased encounters may enable them to secure more exclusive access to the highest-quality areas through the exclusion of smaller groups. While direct evidence is lacking for whether larger groups gain a net advantage from the wet–dry-season transition, our results underscore how short-term interactions can have long-lasting ecological consequences on space use and access to resources.

These findings parallel predictions from the resource dispersion hypothesis (RDH), originally developed to explain group formation in territorial social carnivores[39,60]. In these species, individuals that might otherwise occupy separate territories form groups when resources are sufficiently rich and dispersed to support multiple individuals within a single territory[17,61]. This can promote sociality even in species with little cooperative behaviour or predator-defence needs[34]. Although social primates such as capuchins differ in many respects— including more rigid social structures[62], additional drivers of sociality (for example, alloparental care[44], social learning[63]) and generalist diets[64]—our findings suggest that similar ecological dynamics may still emerge. When the environment can support more individuals or groups, capuchins appear to reduce investment in active defence against neighbour intrusion.

The RDH alone cannot explain capuchin sociality as groups remain hostile during intergroup encounters regardless of resource dispersion[51]. Yet, it may help explain how smaller groups persist on the landscape despite overlapping with multiple more powerful neighbours. For instance, our findings indicate that larger groups tend to avoid overlapping with other large groups, perhaps to reduce the risk and energetic cost of conflict with similarly powerful rivals, where outcomes are less predictable[65–67]. This pattern aligns with game-theoretic models of animal contests, which predict that when opponents are evenly matched, mutual assessment and conflict avoidance strategies evolve to reduce costly escalations[68,69]. Such mutual avoidance may inadvertently benefit smaller groups by creating 'buffer zones'— underused areas that smaller groups can exploit while remaining undetected[70]. Comparable patterns are seen with predator–prey dynamics: white-tailed deer (*Odocoileus virginianus*), for example, are more likely to occur in areas where wolf (*Canis lupus*) pack territories overlap[71,72]. Similarly, during large-scale human conflicts, activity often declines near war zone buffer areas, leading to natural reforestation and surges in wildlife populations and biodiversity[73,74]. Our results suggest that when resources are sufficiently abundant and dispersed to support multiple groups within a given area, spatial overlap increases—especially between groups with strongly asymmetric competitive abilities. Such conditions allow dominant groups to access new resources with minimal conflict while enabling smaller groups to persist in underused zones between the ranges of their more powerful neighbours.

Our results show that greater within-group competition with increasing group size does not necessarily lead to higher daily travel costs, even for species that primarily forage on clumped resources. Folivores are known to avoid this trade-off owing to the relatively

widespread and homogeneous nature of their food resources (the so-called folivore paradox)[40,75]. Dietary generalists may achieve the same effect by shifting to more readily available resources such as insects[76]. Beyond such fallback strategies, our findings suggest that larger groups can also alleviate foraging pressure by expanding their range, increasing the diversity of fruiting trees exploited and lowering revisitation frequency. This strategy probably enables access to less-depleted, higher-quality patches, particularly along range peripheries where fruiting trees contain more ripe fruit[77]. Considering physiological constraints on daily energy expenditure[78,79], rotating among foraging areas over longer timescales may be a more efficient strategy than increasing daily travel distance (Supplementary Note 7).

Furthermore, our findings show that climatic extremes (exceptionally severe dry or wet seasons) exacerbate within-group competition for large groups, leading to decreased per capita foraging efficiency. This pattern suggests that extreme climates compound ecological constraints on larger groups. Tropical forests are particularly sensitive to ENSO-driven anomalies[47,80], which have well-documented fitness consequences for mammals (including capuchins). El Niño events that prolong or intensify the dry season increase heat stress and reduce survival and reproductive success[49,81], whereas La Niña events that amplify wet-season rainfall disrupt phenological and arthropod cycles, occasionally triggering famine and widespread mortality[82,83]. Such extremes probably impose additional constraints on large groups, especially if they suppress overall food production so that even the best-quality patches cannot support many consumers.

By contrast, intermediate climatic anomalies (ENSO-linked conditions that counterbalance the typical seasonal cycle) appear to relax these constraints and may even enhance between-group advantages for large groups. During wetter-than-average dry seasons and drier-than-average wet seasons, group size had little to no effect on per capita foraging efficiency, and larger groups tended to occupy higher-quality ranges. One possibility is that such conditions increase spatial heterogeneity in habitat quality (analogous to the intermediate disturbance hypothesis[84]), creating patchier resource landscapes that larger groups can more effectively monopolize—a pattern consistent with broader evidence that climate variability and resource heterogeneity select for enhanced cooperation and resource defence strategies[85–87]. Thus, in highly heterogeneous environments, between-group competition may have had a greater role in the evolution of sociality than often appreciated[7,88]. Future work that explicitly quantifies the distribution of key resources (beyond vegetation greenness) and links group size to fitness would clarify the ecological conditions that favour within- versus between-group competition. Additional insight into the evolution of sociality may come from integrating these trade-offs within a multilevel selection framework[89,90].

We show that the benefits of between-group dominance and the costs of within-group competition vary predictably across changing environments. Such shifting trade-offs may help explain why group sizes vary widely within populations and over time. Small groups can persist by (1) having reduced within-group foraging pressure, (2) rooting themselves spatially to avoid conflict with neighbours and (3) maintaining a competitive advantage within their core area regardless of opposing group size[37]. By contrast, large groups can endure by asserting spatial dominance and monopolizing the most valuable resources which could act as a buffer against intensified within-group competition in the face of resource unpredictability. Nevertheless, this buffering capacity may have limits under particular social or ecological conditions (for example, prolonged climatic extremes), potentially explaining the rare occasions when large groups permanently fission[91]. Unlike 'facultative groupers' (that is, fission–fusion species) that can flexibly adjust group size to local circumstances[31], 'obligate groupers' such as capuchins are bound by the long-term benefits of group living (for example, cooperative breeding, protection from predators or infanticidal conspecifics[92]), which regularly outweigh the costs (for example, within-group

competition, disease transmission)[29]. For such species, permanent fission represents a last resort when groups approach the upper limit of viable size under prevailing conditions[93], with the potential to fundamentally alter the balance of power across the social landscape.

Understanding how group size influences resource competition is inherently complex. Many studies analyse within- and between-group dynamics in isolation, overlooking how they interact and vary with ecological context. Our study addresses this gap by integrating multi-group longitudinal data with a novel, spatially explicit extension of the SRM. This framework quantifies how both focal and neighbouring group traits shape ranging behaviour, revealing whether competitive pressures are specific to certain group pairs or reflect broader, population-level patterns. A remaining obstacle, however, is accounting for unmonitored groups—a common limitation in long-term field studies[23]. Adopting a more holistic 'neighbourhood' perspective—akin to the distinction between local and global dispersal[94,95]—could offer deeper insight by integrating the combined effects of multiple neighbours and inferring impact of unobserved groups. Our framework provides the foundation for such extensions, and for a more nuanced consideration of underlying social dynamics (for example, age–sex structure, dominance hierarchies, collective action problems[96]).

Our study reveals that climate and demographic change shape the trade-offs linking within- and between-group competition. While within-group competition imposes an upper limit on group size[9], and between-group competition and predation probably set a lower limit[10], the strength of these constraints is context dependent. Their relative influence shifts with climate and resource availability[38], allowing groups of different sizes to coexist, each benefiting from distinct ecological and social conditions. Yet, increasingly intense and erratic ENSO cycles, together with ongoing habitat fragmentation, are rapidly altering tropical landscapes[80,97–99], making resource availability increasingly unpredictable. Such climatic fluctuations appear to amplify within-group competition under some ecological contexts (for example, El Niño dry seasons) while enhancing between-group advantages under others (for example, La Niña dry seasons). These findings raise the question of whether ongoing climate change will destabilize these trade-offs, ultimately tilting the balance in favour of smaller or larger groups in the future.

## Methods

### Ethics statement
The study was purely observational, with GPS units carried by observers rather than attached to animals. Research protocols were approved by the Animal Care Committee of the University of California-Los Angeles (protocol 2016-022), and all required permits from SINAC and MINAE (the Costa Rican agencies overseeing wildlife research) were secured and renewed every 6 months. The most recent authorizations include scientific passport number 012-2024-ACAT and permit Resolución number M-P-SINAC-PNI-ACAT-0010-2024. All procedures complied with the Animal Behavior Society's Guidelines for the Use of Animals in Research[100].

### Data collection
The Lomas Barbudal Monkey Project dataset contains longitudinal records from 13 habituated groups of white-faced capuchins, spanning 1990 to 2025. These records include data on demographics, foraging, social interactions and group movement. The 13 study groups have varying observation periods, reflecting differences in when each group was introduced into the study or formed through permanent fission from an existing study group (Fig. 1c). When a permanent fission occurs, the group that retains the natal home range (invariably the larger faction) keeps the original identifier of the parent group, whereas the group that splits off and forms a new home range is given a new identifier. Here we draw on data from 12 of the 13 study groups, as one group (dt) was excluded owing to insufficient location data.

Location data from 1991 to September 2009 (none were available for 1990) consist of georeferenced sleep-site records digitized from field notes originally used to relocate groups each morning. From September 2009 to 2025, handheld Global Positioning System (GPS) units (Garmin GPSmap 62s, 64, 64s, 66sr) were clipped to or placed in researchers' backpacks and used to record both sleep-site locations and daily movement trajectories as researchers followed the monkeys. GPS tracking data have been cleaned and processed through 2020, and GPS-derived sleep-site locations through 2023. Sleep-site data therefore span from the start of 1991 to the end of 2023, whereas GPS tracking data extend from September 2009 to April 2020 (Fig. 1). Previous work has validated the reliability of both data types for estimating home ranges[43,101].

Researchers positioned themselves near the group's centre when not conducting focal follows. GPS units recorded locations at intervals of 5 min (2009–2012) or 30 s (2013–2020). We calculated median annual and seasonal group sizes from 1991 to 2023 using daily group censuses with at least 6 h of observation to maximize the likelihood of detecting all individuals present. We summarized fruit foraging behaviour from focal follows conducted from 1 July 2006 to 30 June 2021. These records included the number of bites of fruit observed to be ingested, the duration each individual was visible, and the age and sex of the focal individual. Further specifics on data collection and processing are provided in ref. 101 and ref. 102.

## Environmental data

The tropical dry forests in and around the Lomas Barbudal Biological Reserve experience strong annual and seasonal variation in vegetation phenology, affecting the availability of key resources such as food, water and shade for capuchins. To quantify interannual and interseasonal variation in vegetation phenology across the study period, we extracted annual and seasonal NDVI values from 1991 to 2023 using satellite imagery extracted from Google Earth Engine[103]. Although NDVI can be an imperfect proxy for food availability in tropical forests[104], it has been shown to measure long-term habitat variability well in this region[105], particularly in the dry season when resources such as water, fruit and shade are concentrated in greener riparian zones. NDVI quantifies the 'greenness' of each pixel in a given image, serving as a proxy for vegetation health and abundance. We defined a bounding box around the Lomas Barbudal Biological Reserve and surrounding forest as the spatial extent for all image processing. We used surface reflectance data from Landsat 5, Landsat 7 and Landsat 8 satellites to obtain red (RED) and near-infra-red (NIR) bands required for NDVI calculation. To remove cloud-contaminated pixels, we applied a mask using the quality assessment (QA_PIXEL) band. NDVI was calculated for each image as:

$$NDVI = \frac{NIR - RED}{NIR + RED} \qquad (1)$$

For each year and season, we created pixel-wise maximum composites to reduce noise from residual cloud contamination and obtain the best-quality pixel value. Seasonal windows were defined as February–April (peak dry season) and September–November (peak wet season). These seasonal and annual median NDVI images were clipped to the study area and exported at 30-m resolution (see Supplementary Note 3 for details). In addition to NDVI, we extracted rasters of tree cover from the Hansen Global Forest Change dataset[106], which were later used to refine home range estimates (see 'Group-level response variables' in Methods). All processing of remote sensing data was conducted using Google Earth Engine's Python API, and rasters were exported as .tif files for further analysis in R[107].

We quantified seasonal severity using the SPEI, which measures anomalies in water balance calculated as precipitation minus potential evapotranspiration—the amount of water lost to evaporation and plant transpiration under existing temperature and radiation conditions. Monthly SPEI values (1990–2023) were obtained from the ERA5-Drought reanalysis dataset[108] for the Lomas Barbudal Biological Reserve and surrounding areas used by our study population. SPEI values were standardized within seasons, such that negative values indicate abnormally dry periods and positive values indicate abnormally wet periods relative to the typical range within that season (January–April = dry; May–December = wet). We used a 6-month accumulation period, which captures broader climatic conditions and aligns most closely with ENSO variability (Supplementary Fig. 5; see Supplementary Note 3.2 for comparison with 1-month SPEI).

## Group-level response variables

For several of our questions of interest, our observation and response variables of interest were properties of individual groups (Table 1).

**Quantifying fruit intake and daily path length.** Per capita fruit intake rates were estimated from focal follow data collected between 1 July 2006 and 30 June 2021. This dataset includes 4,952 h of usable observation time from 335 individually identified focal individuals. For each focal individual per day, we recorded the number of bites of fruit ingested, rather than whole fruits, as many fruits require multiple bites to consume. Intake data were restricted to periods when both the mouth and hands of the individual were visible, ensuring accurate detection of fine-scale foraging behaviour.

To quantify daily energy expenditure, we used GPS tracking data to generate 996 daily path lengths across 11 groups from September 2009 to April 2020, applying the continuous-time speed and distance method previously described[109] using the ctmmR package[110]. This approach estimates daily path length by multiplying the estimated mean speed from a fitted daily continuous-time movement model by the total daily sampling duration. By separating the continuous-time movement process from the discrete-time sampling process, continuous-time speed and distance are less sensitive to sampling rate, path complexity and location error than traditional methods based on totalling step lengths. We included only GPS tracks with at least 11 continuous hours of data per day, corresponding to minimum daylight hours, as capuchins rarely move at night.

**Quantifying home range area, greenness and revisitation rate.** We used sleep-site data from 1991 to 2023 to estimate home range area (validated in ref. 43), enabling us to use the full 33-year dataset, including years pre-dating GPS, and capture the widest range of demographic changes and study groups over time. Our validation study showed that 98% utilization distributions (UDs) from sleep ranges accurately aligned with the conventional 95% UDs from GPS tracking data as capuchins typically refrain from sleeping in the far outskirts of their home range[43]. Based on this, we used 98% UDs to estimate home range area, representing the smallest area with a 95% probability of finding the group considering its total movements.

We calculated 156 annual and 224 seasonal sleep home ranges across 12 groups using auto-correlated kernel density estimation (AKDE) implemented using the ctmmR package[110]. AKDE estimates the UD by fitting a series of continuous-time movement models to the location data, selecting the best-fitting model based on the data's auto-correlation structure, which informs the kernel smoothing bandwidth and yields home range estimates that account for location error and irregular sampling[111,112]. Seasonal ranges were segmented as January–April (dry) and May–December (wet), based on behavioural shifts in daily distance to rivers, rather than on climatic seasonality, which typically includes December in the dry season (Fig. 1). Only data segments suitable for home range estimation, identified via visual inspection of variogram regression plots, were included in downstream analyses[113]. We also fit these home ranges using an integrated resource selection function of tree cover following a previous study[114]. This imposed 'soft barriers' by downweighting areas without trees, such as pasturelands,

which home range estimates may expand into but capuchins rarely use. We measured home range greenness by extracting the mean NDVI value within each group's mean home range boundary for the corresponding year or season using the `terra` package[115].

We quantified revisitation rates using the `revisitation()` function in the `ctmm` package, which provides a continuous-time, model-based estimate of how frequently a capuchin group returns to previously used areas. After fitting a continuous-velocity movement model and estimating each group's AKDE UD, `revisitation()` constructs a revisitation-rate surface on the same spatial grid as the UD. At each grid cell, the local revisitation rate is determined by the product of the model-estimated instantaneous speed and the UD value at that location. This represents the expected frequency with which the animal's path re-enters a small circular neighbourhood around that point, scaled by the grid's radial resolution. Averaging this revisitation-rate surface over the UD yields a single mean revisitation rate per radial distance for each group, which describes the typical frequency with which a group revisits locations within its home range (that is, how often the group returns to previously used areas versus moving into less-used areas). The revisitation rate is defined per unit metre by default, but can be multiplied by any larger distance (for example, 25–50 m) to obtain an expected revisitation rate for that spatial scale. Because `ctmm` derives revisitation from a continuous-time model rather than from discrete visit-and-leave thresholds, the estimate is insensitive to the choice of revisitation radius and robust to the sampling schedule of the location data (provided the data resolve the movement's velocity timescale). We used the GPS tracking dataset (September 2009–April 2020) to estimate revisitation rates, which were calculated separately for each group-year and group-season, matching the observational scale used for UD estimation. Although narrower in longitudinal scope than our sleep-site records (1990–2023), these tracking data provide the spatial and temporal resolution needed to estimate velocity, generate accurate UDs[43] and capture diurnal space-use patterns relevant to revisitation. An empirical comparison with trajectory-based revisitation estimates from the `recurse` package[116] is provided in Supplementary Note 8.

### Dyad-level response variables

For other questions of interest, our observation and response variables of interest were properties of dyads of groups—a unique pairing of groups (Table 1). This distinction between group- and group-dyad-level outcomes is important for accurate statistical estimation.

**Quantifying proportional overlap and range shifts.** We measured home range overlap between group-dyads among known habituated groups throughout the study period. We calculated the proportion of a group's home range that overlaps with a specific neighbour ($PO_{fn}$). This approach allows us to assess the degree to which a neighbour can encroach into a focal group's range, and the potential loss of a group's available resources to that neighbour. $PO_{fn}$ was calculated using the following equation:

$$PO_{fn} = \frac{IA_{fn}}{HRA_f} \qquad (2)$$

Here $IA_{fn}$ represents the area of intersection between the focal and neighbour group's home ranges, and $HRA_f$ is the home range area of the focal group. $PO_{fn}$ measures an asymmetric group-dyad outcome. To demonstrate, imagine that group A and group B have home ranges with some intersection, $IA_{ab}$. While $IA_{ab} = IA_{ba}$, $PO_{ab} \neq PO_{ba}$ because $HRA_a \neq HRA_b$. Using this asymmetric variable, we focus on the proportional overlap from the focal group's perspective. For every observation, the neighbouring group also has a $PO_{fn}$ measurement but the groups are switched.

While $PO_{fn}$ captures asymmetric space use and can suggest which group is being encroached upon at a given time, it does not track the directionality of range shifts over time. To assess whether larger or smaller groups drove increases in overlap, we conducted a complementary analysis on dyads that showed both a substantial increase in spatial overlap (from <25% to >45%) and a clear change in relative group size (absolute difference ≥5 individuals). These thresholds were chosen to isolate cases with meaningful shifts in both space use and relative group size, avoiding ambiguity in assigning which group is considered the 'big' group (for example, a group may remain larger overall but shrink more than its neighbour from time $t$ to time $t + i$) and minimizing noise from minor fluctuations. For each of these dyads (63 cross-year comparisons across 11 unique group pairs), we measured the distance from each group's home range centroid at time $t$ to the centroid of the overlap zone at both time $t$ and time $t + i$, and calculated the change in distance ($\Delta D$):

$$\Delta D = D_{t+i} - D_t \qquad (3)$$

A positive $\Delta D$ indicates that the group shifted towards the other group in the dyad, while a negative value indicates that the other group shifted towards them. The group with the larger magnitude of $\Delta D$ was considered the one that drove the increase in overlap.

**Quantifying encounter rates.** We computed encounter rates using the `ctmm::encounter()` function, following ref. 56, to quantify each dyad's expected space–time overlap. As with revisitation rates, we used UDs derived from GPS tracking data (2009–2020) to better capture fine-scale, diurnal space use. Importantly, `ctmm::encounter()` does not count empirical instances in which groups simultaneously enter a defined encounter radius. Rather, it estimates how often two groups' modelled space use would place them within a small spatial radius at the same moment, given their long-term movement patterns and assuming stationary movement parameters (for example, range crossing time and autocorrelation structure). Using this framework, we define the encounter rate $ER_{ij}$ as the relative probability that, at any given moment, groups $i$ and $j$ are found within the same 1 $m^2$ area (a scale that can be adjusted to any encounter radius). These values are relative and interpretable by comparing across dyads; for example, if $ER_{ab} = 0.4$ and $ER_{ac} = 0.2$, then groups $a$ and $b$ are twice as likely to encounter each other as groups $a$ and $c$. Unlike $PO_{fn}$, $ER_{ij}$ is symmetric, yielding the same value regardless of which group is labelled $i$ or $j$, as group identity has no directional role in this metric. We also validated that these model-based encounter rates correspond closely with observed intergroup encounters (Supplementary Note 8).

### Statistical analyses

**Group-level models.** We used a suite of GLMMs to examine the relationship between group size and several within-group outcomes: per capita fruit intake rate, daily path length, revisitation rate, home range area and mean home range NDVI. All models were fit in a Bayesian framework using the `brms` package[117]. We specified gamma likelihoods for all models except the home range NDVI models, which used a beta distribution, and the fruit intake models, which used a negative binomial distribution. Group size was standardized, and additional covariates were included where appropriate based on causal assumptions outlined in our directed acyclic graphs (Supplementary Note 4). All models included varying effects by group, with additional varying effects by individual (for fruit intake) or by year (for home range NDVI), where appropriate. The fruit intake models also included an offset term accounting for observation time (seconds in view). Models for home range area and daily path length included error terms on the response to account for measurement uncertainty. Because key resources regenerate annually[118] but also fluctuate between wet and dry seasons, each model was run in two versions: one including seasonality (via an interaction between group size and season) and one without. While only home range area and NDVI were explicitly

measured at seasonal or annual scales, we refer to the non-seasonal models for all outcomes as 'annual' for consistency. To assess how climatic anomalies modulate these dynamics, the seasonal models for home range NDVI and fruit intake rate (proxies for between- and within-group competition, respectively) were extended to include SPEI as a three-way interaction with group size and season. SPEI was standardized within seasons, allowing us to estimate how deviations from typical wet- or dry-season conditions modify the effects of group size without introducing collinearity between SPEI and season.

**Dyad-level models.** To examine how relative group size shapes between-group interactions, we used an extension of the hierarchical SRM[53]. SRMs separate group-level and dyadic-level variation while modelling their correlation, enabling efficient partial pooling and improving parameter estimation[54]. While traditionally applied to social behaviour among individuals or human households (for example, refs. 119–121), we extend this framework to spatial interactions between animal social groups.

We modelled proportional overlap ($PO_{fn}$) using a zero-augmented (hurdle) SRM implemented in the `rethinking` package[122]. The hurdle component separates the probability of non-zero overlap (Bernoulli distribution) from the degree of overlap when present (beta distribution). Key predictors included focal and neighbour group size (as group-level fixed effects) and their interaction (as a dyad-level fixed effect). Seasonal models included a three-way interaction between season, focal group size and neighbour group size. To account for non-independence across dyads, we included varying effects for focal group ID, neighbour group ID and dyad ID, modelled jointly via a multivariate distribution with a shared correlation structure. This structure allows, for instance, inference about whether a group that overlaps more with one neighbour also overlaps more with others—capturing cross-dyad patterns more flexibly than standard random effects. See Supplementary Note 5 for full statistical notation of our SRMs.

We modelled encounter probabilities using a related set of Gaussian-distributed dyadic models (after log transforming the response), implemented in `brms`. While based on the same dyadic framework, these models used a different varying effects structure than the SRM described above (sensu refs. 119,123), omitting correlated focal-neighbour effects. This was appropriate because encounter rate is a symmetric response—interactions between groups $i$ and $j$ are equivalent regardless of order—making the focal-neighbour distinction neither identifiable nor meaningful. To reflect the symmetrical nature of the data, we included two sources of random variation: a dyad-level effect, capturing dyad-specific tendencies in encounter rates, and a group-level multiple-membership structure, allowing each observation to be jointly attributed to both groups involved. This approach preserves the exchangeability of group identities within dyads and appropriately partitions variance between individual group tendencies and unique dyadic relationships. All predictors were also symmetric: group size difference, overlap zone area and vegetation greenness in the overlap zone (measured by mean NDVI). As with the home range and path length models, we accounted for uncertainty in encounter rates using measurement error modelling. To assess seasonal variation in encounters, we included separate interaction terms between season and each predictor—testing whether the influence of group size asymmetries and habitat characteristics on intergroup encounters varied by season.

We applied weakly regularizing priors to all model parameters. All models showed good convergence ($\hat{R} \approx 1.00$), adequate effective sample sizes and reliable fit based on posterior predictive checks. We summarize parameter estimates using 89% highest posterior density intervals (HPDIs) and report the proportion of the posterior greater than zero (PP > 0), which represents the posterior probability that an effect is positive (values near 0 indicate strong support for a negative effect). Following a previous study[122], we use 89% intervals not as

significance thresholds but as a convention to represent uncertainty while discouraging dichotomous interpretation of results (that is, 'significant' versus 'non-significant'). See Supplementary Note 4 for additional model details and assumptions using a causal inference framework.

## Reporting summary
Further information on research design is available in the Nature Portfolio Reporting Summary linked to this article.

## Data availability
Location data are restricted to protect the precise whereabouts of habituated, threatened primates vulnerable to the pet trade. They are archived in a limited-access Movebank repository (https://www.movebank.org/cms/webapp?gwt_fragment=page%3Dstudies%2Cpath%3Dstudy6448057425) and available upon reasonable request via Movebank to S.E.P. (sperry@anthro.ucla.edu). All demographic and behavioural data are openly available via Dryad at https://doi.org/10.5061/dryad.612jm64j0 (ref. 124).

## Code availability
The code necessary to reproduce our analyses is openly available via Edmond[125].

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

## Acknowledgements

We thank K. Tiedeman and A. Ashbury for feedback on remote sensing and writing, respectively. We thank C. Ross and J. Koster for feedback on model structure and R. Berl for establishing the GPS protocols used for location data collection at Lomas (see Supplementary Note 1 for a list of all contributing field assistants). Thanks to D. Cohen for assistance with maintaining the Lomas Barbudal Monkey Project database. We thank the Costa Rican Park Service (SINAC, Área de Conservación Arenal Tempisque) for permission to work in Lomas Barbudal Biological Reserve and the private landowners who have granted us permission to work on their land (especially Hacienda Pelon, Brin d'Amor and the community of San Ramon de Bagaces). Data collection was funded primarily by grants to S.E.P. from the National Science Foundation (NSF grants BCS-1919649, BCS-1638428, BCS-0613226, BCS-848360, DDIG 1232371 (co-PI I Godoy), 9633991 and SES9870429 and an NSF graduate fellowship), the National Geographic Society (grants 7968-06, 8671-09, 9058-12, 9795-15 and 45176R-18 and a grant with co-PI B Smuts), the L.S.B. Leakey Foundation (nine grants), the Templeton World Charity Foundation, Inc. (grant 0208), two Wenner-Gren grants (one with co-PI I. Godoy), various UCLA Council on Research (COR) grants and internal grants from the University of Michigan, Sigma Xi; 5 years of funding by Max Planck Institute for Evolutionary Anthropology and logistic support from the Wild Capuchin Foundation. B.J.B. supplemented data collection via funds from the American Society of Primatologists, ARCS Foundation and an NSF GRF (grant number 1650042). This work was supported by the Alexander von Humboldt-Stiftung through funds awarded to M.C.C.

## Author contributions

We use the Contributor Roles Taxonomy (CRediT; https://credit.niso.org/) to detail author contributions: conceptualization: O.T.J., B.J.B., M.C.C. and S.E.P.; data curation: O.T.J., S.E.P. and B.J.B.; formal analysis: O.T.J. and B.J.B.; funding acquisition: S.E.P., M.C.C. and B.J.B.; investigation: S.E.P., O.T.J. and B.J.B.; methodology: O.T.J., B.J.B., S.E.P. and G.E.F.; project administration: S.E.P., B.J.B. and O.T.J.; resources: S.E.P., B.J.B. and M.C.C.; software: O.T.J., B.J.B. and G.E.F.; supervision: B.J.B., S.E.P., M.C.C. and G.E.F.; visualization: O.T.J.; writing—original draft: O.T.J., B.J.B. and S.E.P.; and writing—review and editing: O.T.J., S.E.P., B.J.B., G.E.F. and M.C.C.

## Funding

## Competing interests

## Additional information

**Correspondence and requests for materials** should be addressed to Odd T. Jacobson.

# Reporting Summary

## Statistics

For all statistical analyses, confirm that the following items are present in the figure legend, table legend, main text, or Methods section.

| n/a | Confirmed | |
|---|---|---|
| ☐ | ☒ | The exact sample size (*n*) for each experimental group/condition, given as a discrete number and unit of measurement |
| ☐ | ☒ | A statement on whether measurements were taken from distinct samples or whether the same sample was measured repeatedly |
| ☒ | ☐ | The statistical test(s) used AND whether they are one- or two-sided *Only common tests should be described solely by name; describe more complex techniques in the Methods section.* |
| ☐ | ☒ | A description of all covariates tested |
| ☐ | ☒ | A description of any assumptions or corrections, such as tests of normality and adjustment for multiple comparisons |
| ☐ | ☒ | A full description of the statistical parameters including central tendency (e.g. means) or other basic estimates (e.g. regression coefficient) AND variation (e.g. standard deviation) or associated estimates of uncertainty (e.g. confidence intervals) |
| ☒ | ☐ | For null hypothesis testing, the test statistic (e.g. *F*, *t*, *r*) with confidence intervals, effect sizes, degrees of freedom and *P* value noted *Give P values as exact values whenever suitable.* |
| ☐ | ☒ | For Bayesian analysis, information on the choice of priors and Markov chain Monte Carlo settings |
| ☐ | ☒ | For hierarchical and complex designs, identification of the appropriate level for tests and full reporting of outcomes |
| ☐ | ☒ | Estimates of effect sizes (e.g. Cohen's *d*, Pearson's *r*), indicating how they were calculated |

*Our web collection on statistics for biologists contains articles on many of the points above.*

## Software and code

Policy information about availability of computer code

| | |
|---|---|
| Data collection | Observational data were recorded in the field using Terminal Emulator for Android on handheld computers. Movement data were collected with Garmin GPSMap devices (62s, 64, 64s, 66sr) and organized using Garmin BaseCamp (version 4.7.5). Environmental data were obtained via the Google Earth Engine Python API (Spyder 6.0.7, Python 3.11.12, Qt 5.15.15, PyQt5 5.15.11, Windows 10). |
| Data analysis | All analyses were conducted in R (version 4.5.0) using RStudio Server (2024.12.1+563). |

For manuscripts utilizing custom algorithms or software that are central to the research but not yet described in published literature, software must be made available to editors and reviewers. We strongly encourage code deposition in a community repository (e.g. GitHub). See the Nature Portfolio guidelines for submitting code & software for further information.

## Data

Policy information about availability of data

All manuscripts must include a data availability statement. This statement should provide the following information, where applicable:
- Accession codes, unique identifiers, or web links for publicly available datasets
- A description of any restrictions on data availability
- For clinical datasets or third party data, please ensure that the statement adheres to our policy

Location data are restricted to protect the precise whereabouts of habituated, threatened primates vulnerable to the pet trade. They are archived in a limited-access Movebank repository here: https://www.movebank.org/cms/webapp?gwt_fragment=page=studies,path=study6448057425. These can be made available

upon reasonable request to Prof. Susan Perry. All code necessary to reproduce our analyses are openly available on Edmond here: https://doi.org/10.17617/3.GBGJCM. All demographic and behavioral data are openly available on Dryad here: https://doi.org/10.5061/dryad.612jm64j0.

# Research involving human participants, their data, or biological material

Policy information about studies with human participants or human data. See also policy information about sex, gender (identity/presentation), and sexual orientation and race, ethnicity and racism.

| Reporting on sex and gender | N/A |
|---|---|
| Reporting on race, ethnicity, or other socially relevant groupings | N/A |
| Population characteristics | N/A |
| Recruitment | N/A |
| Ethics oversight | N/A |

Note that full information on the approval of the study protocol must also be provided in the manuscript.

# Field-specific reporting

Please select the one below that is the best fit for your research. If you are not sure, read the appropriate sections before making your selection.

☐ Life sciences ☐ Behavioural & social sciences ☒ Ecological, evolutionary & environmental sciences

For a reference copy of the document with all sections, see nature.com/documents/nr-reporting-summary-flat.pdf

# Ecological, evolutionary & environmental sciences study design

All studies must disclose on these points even when the disclosure is negative.

| Study description | Aim: This study examines how environmental fluctuations alter the balance between within- and between-group competition, and how these shifts shape group behavior and intergroup interactions at the population scale. |
|---|---|
| | Study system and data: The analysis is based on 33 years (1991–2023) of behavioral and spatial observations from 12 wild groups of white-faced capuchins (Cebus imitator) monitored by the Lomas Barbudal Monkey Project, Costa Rica, since 1990. |
| | Study design structure: Observational, repeated-measures, hierarchical/multilevel design with (i) group-, individual-, and group-dyad-level effects and (ii) nested time scales (daily, seasonal [dry/wet], annual). Dyadic analyses use a hierarchical social-relations framework (focal, neighbor and group-dyad effects). |
| | Predictors ("treatment" factors) and interactions: (i) group size, (ii) season (dry vs. wet), (iii) vegetation productivity (NDVI), (iv) Standardized Precipitation–Evapotranspiration Index (SPEI), (v) dyadic group size (interaction between focal and neighbor group size). Other interactions include: group size x season (group-level models), group size x season x SPEI (fruit intake and range NDVI group-level models), focal group size x neighbor group size x season (dyadic models). |
| | Responses: (i) Within-group models: (a) per-capita fruit ingestion rate, (b) daily path length, (c) return/revisitation rate, (d) home-range area, (e) home-range quality. (ii) Between-group models: (f) home-range overlap and (g) intergroup encounter rate. |
| | Experimental units and replication: (i) Individual-day (fruit ingestion): daily observations nested within 335 individuals nested within 12 groups (varying intercepts/slopes for individuals and groups). (ii) Group-day (daily path length): 996 daily observations nested within 12 groups (varying intercepts/slopes by group). (iii) Group-year and group-season-year (home-range area/quality, revisitation): repeated measures per group across 33 years and 66 seasons/years. (iv) Dyad-year and dyad-season-year (overlap, encounter rate): repeated measures for group pairs; up to 66 possible dyads across 12 groups. |
| Research sample | The research sample comprises 33 years (1991–2023) of behavioral and spatial observations from 12 wild, habituated groups of white-faced capuchins (Cebus imitator) monitored by the Lomas Barbudal Monkey Project in Costa Rica. We aimed to follow as many groups as feasible while maintaining sufficient detail per group to capture between-group variation and population-level patterns. White-faced capuchins are platyrrhine primates well known for their advanced cognitive capacity, large brain-to-body size ratios, omnivorous diets, extractive foraging, cooperation, coalitionary behavior, long life spans, and slow life histories. They live in cohesive multi-male, multi-female groups of roughly 5–40 individuals (mean ≈ 18.8), spanning ages from 1 month to 34 years in our dataset. The sample is intended to represent the free-ranging white-faced capuchin population inhabiting the tropical dry-forest ecosystem surrounding the Lomas Barbudal Biological Reserve and comparable habitats across the Americas. More broadly, the findings have relevance for understanding between-group conflict, resource partitioning, and other ecological trade-offs faced by social animals across taxa, including humans. |
| | We analyze existing datasets compiled by the project. Behavioral data include focal follows collected between July 2006 and June |

2021, yielding 4,952 hours of usable observations from 335 individually identified capuchins. Spatial data come from handheld GPS devices carried by observers following habituated groups between September 2009 and April 2020, supplemented by historical location records extracted from field notes spanning 1990–2023 (e.g., sleep sites and positions at trail crossings and landmarks). Environmental data were derived from openly available satellite products, including surface reflectance from Landsat 5/7/8 (via Google Earth Engine) and tree cover from the Hansen Global Forest Change dataset. Climate data was retrieved from the ERA5 Climate Reanalysis dataset.

**Sampling strategy**

We analyzed the full longitudinal record available from the Lomas Barbudal Monkey Project, using all years, groups, and individuals that met predefined data-quality criteria. This choice reflects the study's observational nature and our aim to capture the widest feasible range of environmental conditions, demographic variation, and intergroup relationships. Sampling intensity varied depending on funding and staff availability, meaning that only one or two groups could usually be monitored at a time. The choice of which group to follow on a given day depended on ongoing project priorities and on which individuals were most in need of updating their focal follow hours for that month. Behavioral focal follows were collected under established protocols for longitudinal studies involving habituated diurnal animals (Perry 2012; https://doi.org/10.1016/B978-0-12-394288-3.00004-6) and we applied basic quality control measures (e.g., excluding incomplete days and spurious locations) where appropriate.

For metrics derived from continuous-time movement models, sample sufficiency was assessed via diagnostics that measure effective information content rather than raw point counts (Fleming et al. 2019; https://doi.org/10.1111/2041-210X.13270). These diagnostics test whether a data segment (day, season, or year, depending on the analysis) contained enough independent information (typically about ten home-range crossings) to estimate parameters such as home-range size, revisitation, or overlap. Segments that failed range-residency tests or fell below this effective sample size threshold were excluded. We further checked that retained data were temporally representative within each analysis window (Jacobson et al. 2024; https://doi.org/10.1007/s10764-023-00398-z).

For statistical models, we relied on Bayesian generalized linear mixed models that make full use of the study's hierarchical and repeated-measures structure. These models included varying intercepts and slopes at the individual, group, and dyad levels, enabling partial pooling. In practice, this means that when some groups or individuals had fewer observations, their estimates were stabilized by borrowing information from the broader dataset, avoiding biased or unreliable results due to uneven sampling. Model adequacy was confirmed through posterior predictive checks, and precision was ensured by the large number of repeated observations across daily, seasonal, and annual scales combined with the breadth of demographic and environmental variation in the dataset.

**Data collection**

Capuchin groups were followed daily from dawn to dusk (typically 05:00–18:15), as they moved from one sleeping site to the next. Because capuchins show little nocturnal activity, observers left groups at nightfall and returned before first light. At each morning sleep site, observers enabled the tracking function on a handheld Garmin GPS unit, which logged locations at ~30-s intervals until the group settled at its evening tree. On days when groups were first encountered in the forest ("search days"), GPS tracking began upon contact and continued until the group reached its sleeping site. Tracking was occasionally terminated early if groups were lost under difficult conditions or if observers alternated between study groups.

While following the monkeys, trained observers collected behavioral data through focal follows, usually 10 minutes in duration (though sometimes longer depending on concurrent projects). During these follows, every behavior of the focal individual was recorded, including foraging and food-processing actions and social interactions. Between focal follows, group scans were conducted to provide snapshots of visible individuals, their activities, and proximity to other group members. Daily censuses were also completed, documenting group membership and health and reproductive status. All individuals were recognized by experienced observers through morphological characteristics.

Before the introduction of handheld GPS units in late 2009, spatial data (especially sleep sites) were documented in field notebooks or handheld Psion devices. Observers described locations relative to salient landmarks such as rivers, trails, waterfalls, and cliffs. These records were essential for relocating groups each morning and maintaining continuity in long-term behavioral sampling prior to the adoption of GPS technology (see Jacobson et al. 2024; https://doi.org/10.1111/ele.14443).

In addition to Susan Perry, Odd Jacobson, and Brendan Barrett, the following field assistants contributed with data collection: J. Anderson, C. Angyal, L. Appleby, K. Atkins, A. Autor C., M. Bergstrom, R. Berl, L. Beaudrot, T. Bishop, A. Bjorkmann, L. Blankenship, T. Borcuch, J. Broesch, A. Büry, D. Bush, J. Butler, F. Campos, C. Carlson, S. Carnegie, S. Caro, L. Chuaqui, A. Cobden, C. Collins, G. Corradini, M. Corrales, J. Damm, B.A. Davis, C. deRango, C. Dillis, N. Donati, G. Dower, R. Dower, A. Duchesneau, K. Feilen, J. Fenton, S. Fiello, K. Fisher, A. Fuentes J., M. Fuentes, T. Fuentes A., A. Gaston, C. Gault, H. Gilkenson, M. Glenwright, I. Godoy, I. Gottlieb, J. Griciute, J. Gros-Louis, L.M. Guevara R., M. Guimond, L. Hack, M. Hammel, R. Hammond, R. Hamrick, A. Hanadari-Levy, S. Herbert, C. Hirsch, M. Hoffman, C. Holman, J. Hubbard, S. Hyde, M. Jackson, S. Jackson, E. Johnson, K. Kajokaite, M. Kay, E. Kennedy, D. Kerhoas-Essens, S. Kessler, D. Khieninson, P. Kolence, W. Krimmel, W. Lammers, M. Lechner, S. Lee, S. Leinwand, L. Johnson, S. Lopez Plaza, T. Lord, S. MacCarter, J. Mackenzie, F. McKibben, J. Manson, M. Mayer, W. Meno, A. Mensing, M. Milstein, A. Mitchell, C. Mitchell, W. Meno, J. Mudde, Y. Namba, D. Negru, A. Neyer, C. O'Connell, J.C. Ordoñez J., F. Ouweleen, N. Parker, B. Pav, S. Pereira, K. Perry, J. Pinnock, R. Popa, K. Potter, K. Ratliff, K. Reinhardt, N. Roberts B., E. Rothwell, J. Rottman, H. Ruffler, S. Sanford, C.M. Saul, S. Schading, I. Schamberg, S. Schembari, N. Schleissmann, K. Schleper, C. Schmitt, S. Schulze, A. Scott, E. Seabright, J. Shih, L. Sirot, S. Sita, M. Skuja, J. Stampfl, K. Stewart, W.C. Tucker, E. Urquhart, J. Vandermeer, K. van Atta, L. van Zuidam, J. Verge, G. Viallon, V. Vonau, R. Wakeford, A. Walker-Bolton, K. Watz, E. Wikberg, M. White, E. Williams, J. Williams, E. Wolf, D. Wood, D. Works, and M. Ziegler. Long-term site managers H. Gilkenson and W. Lammers made particularly large contributions.

**Timing and spatial scale**

Data collection for this study spans from 18 May 1991 to 24 September 2023. Because field staff were limited and groups were followed on a rotating basis, sampling effort was uneven across groups and years, leading to data gaps for particular groups. These gaps and their implications for spatial analyses were evaluated in detail by Jacobson et al. (2023; https://doi.org/10.1007/s10764-023-00398-z). That analysis demonstrated that missing data did not bias home-range estimates provided that observations were distributed across at least 5–10 unique weeks within a given season or year. Following those results, we included only data segments that met or exceeded these thresholds.

Sampling frequency and periodicity varied: groups were observed for runs of 1–22 consecutive days, with focal follows and GPS

tracking conducted continuously during daylight hours. Exact sampling periods for each group are shown in Supplementary Figure S1. The spatial scale of the study site encompasses approximately 10 km² of tropical dry forest within and adjacent to the Lomas Barbudal Biological Reserve, Costa Rica, which includes the overlapping home ranges of all monitored groups.

**Data exclusions**

Data segments were excluded if they did not meet pre-established thresholds for sample size sufficiency. Most notably, data from 1990 were too sparse and therefore excluded, and one group (DT) was removed due to insufficient location data; thus, although 13 groups were habituated, only 12 are included in the analyses. In addition, GPS tracking data collected after April 2020 were not yet fully cleaned and processed, so only GPS data from September 2009–April 2020 were included in the analyses. Sleep-site data, however, span continuously from 1991 through September 2023. All exclusions were based on predefined criteria established in previous validation work (Jacobson et al. 2023; Jacobson et al. 2024).

**Reproducibility**

Our study is observational and does not involve experiments, but all analyses are fully reproducible. Code, workflows, and documentation are openly available at https://doi.org/10.17617/3.GBGJCM, with detailed annotations. No attempts at replication have failed; the analyses can be rerun to yield the same results.

**Randomization**

Our study did not involve randomized experiments, as it is based on long-term observational data. Instead, we controlled for covariates statistically using a causal inference framework. For each model, we developed directed acyclic graphs (DAGs) to make our causal assumptions explicit and to guide covariate selection. The rationale and details of this approach are provided in Section S1.4 of the Supplementary Material.

**Blinding**

Blinding was not relevant to this study because it was purely observational (no treatment groups). All groups of capuchins are natural social units that exist in the wild, and all individuals are identified by trained observers. Rigorous training and regular examinations ensured that all data was collected reliably and to the highest standard.

Did the study involve field work?   ☒ Yes   ☐ No

# Field work, collection and transport

**Field conditions**

The study was conducted in tropical dry forest in Guanacaste, Costa Rica, within and around the Lomas Barbudal Biological Reserve. The climate is strongly seasonal, with a dry season (December–April) and a wet season (May–November). During the dry season, daily maximum temperatures average 30–35 °C with near-zero rainfall, while in the wet season temperatures average 27–32 °C and daily precipitation ranges from ~5 to >300 mm (Figure 1; ERA5). The terrain is moderately hilly and includes continuous forest inside the reserve as well as fragmented pasture–forest mosaics with riparian corridors outside. Rivers are a regular feature of the landscape and were often crossed either while following monkeys or when traveling between sleeping sites and vehicles. The reserve is also notable for high bee and wasp diversity and the presence of venomous snakes (e.g., rattlesnakes); field teams followed standard safety protocols, including avoiding wearing snake leggings and maintaining contact with park rangers. For much of the study, observers resided in a nearby town (~30 min by vehicle) and commuted daily, arriving before dawn and returning after dusk; at other times they lived inside the reserve and walked directly to sleep sites.

**Location**

Within and nearby the Lomas Barbudal Biological Reserve in Guanacaste, Costa Rica (10∘29–32'N, 85∘21–24'W). The elevation is around 10-180m (Frankie et al. 1993; https://doi.org/10.2307/2388320).

**Access & import/export**

We obtained permission from the Costa Rican Park Service (SINAC, Área de Conservación Arenal Tempisque) to conduct research in the Lomas Barbudal Biological Reserve, as well as consent from private landowners who granted us access to their properties, including Hacienda Pelón, Brin d'Amor, and the community of San Ramón de Bagaces. The study was purely observational: no animals were captured or handled, no invasive procedures were performed, and no samples were imported or exported. Research protocols were approved by UCLA's Animal Care Committee (protocol 2016–2022), and all required permits from SINAC and MINAE (the Costa Rican agencies overseeing wildlife research) were secured and renewed every six months. The most recent authorizations include scientific passport #1012-2024-ACAT and permit Resolución #M-P-SINAC-PNI-ACAT-0010-2024. All procedures complied with the Animal Behavior Society's Guidelines for the Use of Animals in Research (https://doi.org/10.1016/S0003-3472(23)00317-2).

**Disturbance**

Our study was strictly noninvasive: animals were not captured, handled, or provisioned, and observers did not interact directly with them. The primary potential disturbance arises from habituation of capuchins to researcher presence, which may alter aspects of their behavior and reduce interactions with predators or competitors. This is an unavoidable outcome of all individual-based longitudinal studies on wild primates.

# Reporting for specific materials, systems and methods

We require information from authors about some types of materials, experimental systems and methods used in many studies. Here, indicate whether each material, system or method listed is relevant to your study. If you are not sure if a list item applies to your research, read the appropriate section before selecting a response.

## Materials & experimental systems

| n/a | Involved in the study |
|---|---|
| ☐ ☐ | Antibodies |
| ☐ ☐ | Eukaryotic cell lines |
| ☐ ☐ | Palaeontology and archaeology |
| ☐ ☒ | Animals and other organisms |
| ☐ ☐ | Clinical data |
| ☐ ☐ | Dual use research of concern |
| ☐ ☐ | Plants |

## Methods

| n/a | Involved in the study |
|---|---|
| ☐ ☐ | ChIP-seq |
| ☐ ☐ | Flow cytometry |
| ☐ ☐ | MRI-based neuroimaging |

# Antibodies

| Antibodies used | N/A |
|---|---|
| Validation | N/A |

# Eukaryotic cell lines

Policy information about cell lines and Sex and Gender in Research

| Cell line source(s) | N/A |
|---|---|
| Authentication | N/A |
| Mycoplasma contamination | N/A |
| Commonly misidentified lines (See ICLAC register) | N/A |

# Palaeontology and Archaeology

| Specimen provenance | N/A |
|---|---|
| Specimen deposition | N/A |
| Dating methods | N/A |

☐ Tick this box to confirm that the raw and calibrated dates are available in the paper or in Supplementary Information.

| Ethics oversight | N/A |
|---|---|

Note that full information on the approval of the study protocol must also be provided in the manuscript.

# Animals and other research organisms

Policy information about studies involving animals; ARRIVE guidelines recommended for reporting animal research, and Sex and Gender in Research

| Laboratory animals | N/A |
|---|---|
| Wild animals | This study did not involve capturing, handling, or killing any animals. All research was purely observational. The study species was the white-faced capuchin (Cebus imitator), with observed individuals ranging in age from 1 day to 34 years (and possibly older in cases where exact birth dates were unknown). |
| Reporting on sex | Findings apply to both female and male capuchins. In our models evaluating the effect of group size and seasonality on per capita fruit ingestion rate, we included a covariate indicating the sex of the individual in the model. Out of 336 individuals, 183 were male, 149 were female, and 4 were unknown. The data-set used for this analysis will be openly available after acceptance at the Dryad repository: https://doi.org/10.5061/dryad.612jm64j0. |
| Field-collected samples | No samples were collected that are relevant for this study. |
| Ethics oversight | Research protocols were approved by UCLA's Animal Care Committee (protocol 2016–2022), and all required permits from SINAC and MINAE (the Costa Rican agencies overseeing wildlife research) were secured and renewed every six months. The most recent authorizations include scientific passport #1012-2024-ACAT and permit Resolución #M-P-SINAC-PNI-ACAT-0010-2024.All procedures |

complied with the Animal Behavior Society's Guidelines for the Use of Animals in Research (https://doi.org/10.1016/S0003-3472(23)00317-2).

Note that full information on the approval of the study protocol must also be provided in the manuscript.

# Clinical data

Policy information about clinical studies

All manuscripts should comply with the ICMJE guidelines for publication of clinical research and a completed CONSORT checklist must be included with all submissions.

| | |
|---|---|
| Clinical trial registration | N/A |
| Study protocol | N/A |
| Data collection | N/A |
| Outcomes | N/A |

# Dual use research of concern

Policy information about dual use research of concern

## Hazards

Could the accidental, deliberate or reckless misuse of agents or technologies generated in the work, or the application of information presented in the manuscript, pose a threat to:

No | Yes
☒ ☐ Public health
☒ ☐ National security
☒ ☐ Crops and/or livestock
☒ ☐ Ecosystems
☒ ☐ Any other significant area

## Experiments of concern

Does the work involve any of these experiments of concern:

No | Yes
☒ ☐ Demonstrate how to render a vaccine ineffective
☒ ☐ Confer resistance to therapeutically useful antibiotics or antiviral agents
☒ ☐ Enhance the virulence of a pathogen or render a nonpathogen virulent
☒ ☐ Increase transmissibility of a pathogen
☒ ☐ Alter the host range of a pathogen
☒ ☐ Enable evasion of diagnostic/detection modalities
☒ ☐ Enable the weaponization of a biological agent or toxin
☒ ☐ Any other potentially harmful combination of experiments and agents

# Plants

| | |
|---|---|
| Seed stocks | N/A |
| Novel plant genotypes | N/A |
| Authentication | N/A |

# ChIP-seq

## Data deposition

☐ Confirm that both raw and final processed data have been deposited in a public database such as GEO.

☐ Confirm that you have deposited or provided access to graph files (e.g. BED files) for the called peaks.

Data access links
*May remain private before publication.*
> N/A

Files in database submission
> N/A

Genome browser session
(e.g. UCSC)
> N/A

## Methodology

Replicates | N/A
Sequencing depth | N/A
Antibodies | N/A
Peak calling parameters | N/A
Data quality | N/A
Software | N/A

# Flow Cytometry

## Plots

Confirm that:

☐ The axis labels state the marker and fluorochrome used (e.g. CD4-FITC).

☐ The axis scales are clearly visible. Include numbers along axes only for bottom left plot of group (a 'group' is an analysis of identical markers).

☐ All plots are contour plots with outliers or pseudocolor plots.

☐ A numerical value for number of cells or percentage (with statistics) is provided.

## Methodology

Sample preparation | N/A
Instrument | N/A
Software | N/A
Cell population abundance | N/A
Gating strategy | N/A

☐ Tick this box to confirm that a figure exemplifying the gating strategy is provided in the Supplementary Information.

# Magnetic resonance imaging

## Experimental design

Design type | N/A
Design specifications | N/A
Behavioral performance measures | N/A

## Acquisition

| | |
|---|---|
| Imaging type(s) | N/A |
| Field strength | N/A |
| Sequence & imaging parameters | N/A |
| Area of acquisition | N/A |

Diffusion MRI ☐ Used ☐ Not used

## Preprocessing

| | |
|---|---|
| Preprocessing software | N/A |
| Normalization | N/A |
| Normalization template | N/A |
| Noise and artifact removal | N/A |
| Volume censoring | N/A |

## Statistical modeling & inference

| | |
|---|---|
| Model type and settings | N/A |
| Effect(s) tested | N/A |

Specify type of analysis: ☐ Whole brain ☐ ROI-based ☐ Both

| | |
|---|---|
| Statistic type for inference | N/A |

(See Eklund et al. 2016)

| | |
|---|---|
| Correction | N/A |

## Models & analysis

| n/a | Involved in the study |
|---|---|
| ☐ | ☐ Functional and/or effective connectivity |
| ☐ | ☐ Graph analysis |
| ☐ | ☐ Multivariate modeling or predictive analysis |

| | |
|---|---|
| Functional and/or effective connectivity | N/A |
| Graph analysis | N/A |
| Multivariate modeling and predictive analysis | N/A |

nature portfolio | reporting summary

April 2023

8