## [Peer Review File · Nature Ecology & Evolution]

Environmental fluctuations alter the competitive trade-offs of group size in a social primate

Corresponding Author: Dr Odd Jacobson

Version 0:

Decision Letter:

14th October 2025

Dear Dr Jacobson,

Your manuscript entitled "Between-group competitive advantage offsets foraging costs for bigger groups in harsher seasons" has now been seen by three reviewers, whose comments are attached. The reviewers have raised a number of concerns which will need to be addressed before we can offer publication in Nature Ecology & Evolution. We will therefore need to see your responses to the criticisms raised and to some editorial concerns, along with a revised manuscript, before we can reach a final decision regarding publication.

In addition to the general reviewer comments, please pay special attention to the accessibility of the code files--two of the reviewers were unable to access them during this round.

We therefore invite you to revise your manuscript taking into account all reviewer and editor comments. Please highlight all changes in the manuscript text file [OPTIONAL: in Microsoft Word format].

* If you have not done so already please begin to revise your manuscript so that it conforms to our Article format instructions at <http://www.nature.com/natecolevol/info/final-submission>. Refer also to any guidelines provided in this letter.

* Extended Data Figures - please ensure that any supplementary figures and tables that are crucial to the manuscript's conclusions are converted into Extended Data figures and tables to increase visibility of these data. Extended Data figures and tables are online-only (present in the online PDF and full-text HTML versions of the paper), peer-reviewed display items that provide essential background to the article but are not included in the main article due to space constraints. A maximum of ten Extended Data display items (figures and tables) is permitted.

Link Redacted

Nature Ecology & Evolution is committed to improving transparency in authorship. As part of our efforts in this direction, we are now requesting that all authors identified as 'corresponding author' on published papers create and link their Open Researcher and Contributor Identifier (ORCID) with their account on the Manuscript Tracking System (MTS), prior to acceptance. ORCID helps the scientific community achieve unambiguous attribution of all scholarly contributions. You can create and link your ORCID from the home page of the MTS by clicking on 'Modify my Springer Nature account'. For more information please visit www.springernature.com/orcid.

[redacted]

Reviewer expertise:

Reviewer #1: inter/intra group dynamics in primates

Reviewer #2: primatology, social complexity, energetics

Reviewer #3: spatial ecology, animal movement, remote sensing

Reviewers' comments:

Reviewer #1 (Remarks to the Author):

This manuscript represents an excellent endeavor to leverage the power of a long-term dataset that included broad cross-sectional sampling, as well, to tease apart interrelated and confounding effects of within- and between-group competition. I'm unaware of any prior attempts to tackle similar analyses, and indeed I'm not certain there are many datasets available that could attempt to tackle this question with a similar time depth and cross-sectional breadth. I am, therefore, both very excited by the potential that the manuscript represents and inspired by the approach, but I have a few issues that I think may need to be addressed before the results presented fully track with the authors' interpretation. Most critically, I think there is a bit of an overreach on the relative benefits of success in between-group contest competition and the costs of increased within-group contest and scramble, particularly as the authors then extrapolate to potential effects of an increasingly extreme climate within the site. I've noted below my concerns about the interpretations, all of which I think can be addressed through the inclusion of additional analyses and/or increased clarity of the text, if the case is that I am misinterpreting the results.

Major comments:

Lines 22-24: The first sentence suggests primacy of the benefits in between-group competition as a benefit of large group size, disregarding the impact of group size on predation risk. Overall, it feels that this potential benefit of group-living is given very little acknowledgement throughout the text. This may reflect the authors' perceptions of the relative importance of BGC vs. predation-sensitive behavioral strategies, but it does not feel fully representative of the broad literature on the topic.

Lines 112-117: For reference are there substantial increases that occurred in the absence of shifts in relative group size? Also, given the example in Figure 4 shows an 8-year step between the two samples, it appears that the "+1" component of $t+1$ is actually variable (might this be better called " $t+i$ "?). I understand that this may be necessary because of sampling distribution, but it may mean that the timing of the shift in relative group size and the timing of the encroachment are not as tightly correlated as suggested in the text.

Lines 142-144: I'm hesitant about saying that decreased home range overlap suggests territoriality

Lines 146-149: Again just because frequent encounters lead to spatial separation does not mean that the mechanism is territoriality or increased IGE costs for subordinate groups. Missing in this argument is evidence that the proportional shift during the dry season is greater for smaller groups, rather combining Lines 88-90 and lines 110-111 suggest that this is not the case; proportional home range overlap does not increase for small groups, and it's not just because they're contracting their ranges while large groups occupy the area. This finding can also be interpreted then as a decision by groups to avoid foraging in areas that are potentially already depleted. Frequent encounters could occur because groups are forced to revisit the area, but that they don't continue closer to the center of the range of the other because an encounter suggests that that BGS may be high. Can you repeat the annual analysis of directional range shift using the wet-dry season transition to get at this idea of whether large and small groups experience pressure from the neighbor unequally?

Lines 198-201: This interpretation appears inconsistent with the data presented. Regardless of whether there is a difference between wet and dry season in the costs of larger groups size, it does appear to exist. Lower intake in large groups is not

balanced by reduced travel costs, so one would predict lower average net energy gain in large groups. This does not therefore suggest that benefits from between-group contest competition outweigh the costs of within-group contest competition during the dry season, but that leveraging the greater competitive ability of the group means that they do not experience exacerbated costs of within-group contest competition during these seasons because they can gain access to the higher quality areas. Energetically they're still at a net negative relative to the smaller group.

Lines 238-240: This last point seems to be exceeding the data, as well. Again, during the dry season it does not appear that large groups are doing better than small groups, they're just not doing any worse relatively than they are in the wet season. Could you run an additional analysis including the severity of the dry season to get at this point?

Minor comments:

Line 61: Should provide whether this is percent time or based on amount consumed.

Figure S3b: The key to the colors seems to be based on mean NDVI (same as represented in the y-axis), but the y-values where colors change are inconsistent. I think this means that I'm misreading the table, but I cannot come up with another value to base this on from the figure legend.

Line 893-894: This assumption seems out of keeping with territoriality in some species wherein lower overlap is maintained specifically through frequent interactions at borders.

Reviewer #1 (Remarks on code availability):

Attempts to review the data came up with "the requested file is not found".

Reviewer #2 (Remarks to the Author):

This study examines how environmental and demographic factors influence competitive interactions within and between primate social groups. Using a remarkable multi-year and multi-group dataset, the authors approach long-standing questions regarding the evolution of sociality with the latest in analytical techniques. The contributions of this work will be significant and I believe hold tremendous interest to the readership of this journal. Specific strengths include the exceptional clarity of writing, the well-coordinated analyses which add phenomenal robustness to their findings, and the uniqueness of their dataset. My critiques and questions are minimal. First, I wonder if the authors could succinctly address the distinction between within-group contest competition vs. within-group scramble competition. If the authors are perhaps choosing to focus solely on within-group scramble competition, it would be helpful to state this explicitly. Second, my understanding of Figure 1c raises the question of group id following a permanent fission (or fusion). It appears that, upon group fission, one of the fission products retained the former group's name. If this is correct, how did the authors determine which fission product retained the former group's name and what are the implications of this group-level analyses? Lastly, and most minor, with regard to Table 1: I may be misunderstanding a fairly minor point, but how does proportional overlap provide insight into encroachment by neighbor as opposed to encroaching into a neighbor's space? My understanding is that these would be impossible to disentangle in the data available.

Reviewer #2 (Remarks on code availability):

Please see above.

Reviewer #3 (Remarks to the Author):

Overall assessment

This submission presents a beautiful case study of the Ecological Constraints Model of group size, showcasing how within- and between-group competition interact in shaping population-level patterns of space use in relation to group size and environmental seasonality. The dataset, spanning more than 3 decades of field-observations on 10+ neighbouring groups of a wild and free-ranging primate, is nothing short of spectacular, and the authors do it justice by the -overall- thoughtful application of state-of-the-art (Bayesian) linear models. The manuscript itself is clearly organized, has a compelling logical structure, and is very well-written. I further commend the authors for their exemplary supplementary materials that indeed appear to allow the reproduction of all statistical, and conceptual, analyses. In short, there is a lot here that warrants recommendation for publication in *Nature Ecology & Evolution*! I have taken the liberty below, however, to offer some critical suggestions to further strengthen this MS.

Minor concerns

I thought the use of the term "demography" as synonymous with group size throughout the manuscript was a little misleading. No consideration is given to other key demographic traits such as group structure in terms of relatedness, or

age/sex composition, for example. If, for whatever justifiable reason, it was decided to restrict the analyses to group size only, that is totally fine, just don't refer to it as demography then.

The presentation of within- and between-group competition felt a little static on occasion, and important underlying social dynamics such as (the steepness of) within group dominance hierarchies, or coordination problems in mustering effective defence of a home range could (should?) be mentioned in the Introduction and/or discussion.

The most serious 'minor' gripe I have with this study, however, is the idiosyncratic use of 89% HPDIs in the presentation of statistical results. I would rather see more conventional 50% and 95% HPDIs in graphs, for example, as well as the explicit reporting of the total proportion of the posterior probability in line with an effect being in a certain hypothesized direction (as you did in line 91). The authors now very much use 89% HPDIs as arbitrary cut-off values to decide whether they found an effect or not (quite contrary, I imagine, to what McElreath had in mind when introducing "yet another arbitrary" criterion such as 89% CIs). It is further worth pointing out that the majority of reported effects would not have achieved "statistical significance" in the frequentist sense of a two-tailed hypothesis test at $\alpha = 0.05$. In fact, with the use of the 89% HPDI criterion in this study, the two-tailed frequentist equivalent of $\alpha = 0.22$ seems rather (overly?!) generous to make claims about having found an effect or not!

Major concerns

My main reservation with this study, however, concerns the matter that at least 3 of the 7 response variables (daily path length, revisitation rate, and encounter rate) are highly derived, modelled estimates, rather than empirically observed values. This raises two concerns:

1. to varying degrees, I am unsure of the biological information content of these metrics, so this should be expanded upon
2. I can see at least 3 methodological caveats with specifying a modelled estimate as the response variable in a subsequent model. These potential risks need to be addressed more explicitly:

- Error propagation
- Lack of independence
- Double-counting of effects

As to the first point, I genuinely struggle to see, for example, the biological relevance of revisitation rate, calculated as it is here, as a global home range metric of "the mean revisitation rate per radial meter". To me, revisitation rates make biological sense for depletable resources within a home range, not as a proxy of the rate of passing through a typical m² of home range. The interpretation of the calculated encounter rate is equally ambivalent: the 'ctmm' algorithm does not take into account that animals detect (and react to) each other's proximity, well ahead of an encounter. As such, the estimated quantity represents an expectation under some sort of "null-model" of collision rates among inert (but not randomly, because autocorrelated, moving) particles. How this measure informs us about actual encounter rates of biological agents, or how it captures the extent of (experienced) between-group competition, is not clear.

As to the methodological caveats, I recognize that the authors add an error term around some (but not all?) of their modelled response variables when specifying them as the outcome in subsequent (Bayesian linear) models, presumably to alleviate concerns about error propagation? But, I'm not even sure to what extent modelled estimates can indeed be viewed as independent values, given that they are a function of the same underlying data and 'ctmm' model structure. I imagine that reporting on the independence of (Bayesian) model residuals would go a long way to alleviate this concern (on another but related note: wouldn't a multivariate approach, in which the correlation matrix amongst each of the outcome variables is explicitly estimated and incorporated, be preferable over fitting several models in isolation?). Lastly, the secondary (Bayesian linear) models may implicitly re-test patterns (e.g. the autocorrelation term estimated in the 'ctmm' algorithm, which is plausibly driven by group size, or environmental variation) that were explicitly built into the outcome by the value generating process (the ctmm-model), thereby double-counting effects. This may result in biased and/or inflated estimates of effect sizes in the Bayesian models.

Recommendation

I hope that by addressing the concerns expressed above, the authors are able to produce a revised version of their manuscript, that I would -wholeheartedly- be willing to recommend for publication!

Line by line comments

Line 23 -24

"mediated by additional ecological and social factors" Or, alternatively, modulated by local heterogeneity.

Figure 2 (an all similar figures of effect sizes)

I really like that you show the posterior distributions in grey, but would love to see medians along with 50 and 95% HPDIs, rather than -the equally arbitrary, but far less common- 89% HPDIs. You now very much seem to use these 89% values as "critical values", which is not in line with your Bayesian approach.

Line 91

Here you actually report the proportion of the posterior probability that supports your hypothesis ($PP > 0 = 0.94$). I would like to see this reported for all effects!

Line 123

How do these predicted encounter rates relate to observed encounter rates: was there any way to empirically validate these derived estimates?

Line 133
Group demography is not the same as group size!

Line 174 -176
But you didn't find evidence in your study that home ranges of smaller groups are marginal in terms of lower NDVI values.

Line 185 – 187
Is this bit, alluding to advanced cognitive abilities, required? You certainly can't say anything about this on the basis of the data you present in this study, so it reads a little overly speculative.

Line 234
"these constraints are not fixed" I'm not aware of anyone ever claiming they were?

Line 238 -240
Is the link to climate change required here? It's rather speculative to argue that, on the basis of your study, habitat fragmentation should favour larger groups. There are many other reasons why this may or may not be the case.

Line 255 – 256
Nice that you could validate the soundness of this approach: would have been great to see similar validations of your other response variables.

Line 333
What does a single value for an entire home range mean... Are you sure that a similar global value cannot come about through very different local scenarios within a home range?!

Line 348
How did you handle triadic home range overlap? It seems possible, in theory, for a group to have its different dyadic POs to sum up to over 100% with your current definition. Is this problematic?

Line 360
How many of these dyads were there?

Line 372
This definition assumes that groups are impervious to each other's proximity or general whereabouts, a very unrealistic assumption.

Line 373
"These values are relative and only interpretable when compared across dyads". But, are they even interpretable in a biologically meaningful way at all?

Line 377
I wonder whether the dependency between the different outcomes could/should have been taken into account, in an explicitly multivariate model framework for example.

Reviewer #3 (Remarks on code availability):

(see comments under "overall assessment")

*****END*****

Version 1:

Decision Letter:

12th February 2026

Dear Dr. Jacobson,

Thank you for submitting your revised manuscript "Environmental fluctuations alter the competitive trade-offs of group size in a social primate" (NATECOLEVOL-25082592A). It has now been seen again by the original reviewers and their comments are below. The reviewers find that the paper has improved in revision, and therefore we'll be happy in principle to publish it in Nature Ecology & Evolution, pending minor revisions to satisfy the reviewers' final requests and to comply with our editorial and formatting guidelines.

If the current version of your manuscript is in a PDF format, please email us a copy of the file in an editable format (Microsoft

Word or LaTeX)-- we can not proceed with PDFs at this stage.

If you have not done so already, please ensure that you also email us a completed copy of the Reporting summary :

Reporting summary: https://www.nature.com/documents/nr-reporting-summary.pdf

[redacted]

Reviewer #2 (Remarks to the Author):

In my second review of this manuscript, I find the authors have enhanced their study not only with relatively minor text updates but - more significantly - with thoughtfully executed additional analyses focused on severe seasonal variability and model-derived metric validation. These revisions and additional analyses add much to an already strong manuscript. Specifically, several particular yet significant elements of the manuscript (e.g. the concept of territoriality vs. range overlap) have been clarified and the new analyses wonderfully extend the results presented in the first submission. In my opinion, the authors have successfully addressed the reviewer concerns either through manuscript edits or, in some cases, detailed clarifications in their response letter. As noted in my prior review, I believe this study has been superbly designed and executed and that its findings will be of great interest to the journal's readership.

Reviewer #2 (Remarks on code availability):

I continue to have trouble with the Github link, but have reviewed the contents of the posted zip file to the best of my ability.

Reviewer #3 (Remarks to the Author):

Congratulations to the authors on doing an excellent job addressing the concerns raised about the previous version of their manuscript. I particularly appreciate the addition of new analyses to empirically validate the cttm-derived movement metrics: this must have been a lot of work, but I believe really strengthens their study. Although I remain somewhat cautious about the interpretability of a "globally averaged" revisitation rate, I very enthusiastically recommend this paper for publication at this point: a beautiful, and well-executed study!

Reviewer #3 (Remarks on code availability):

I did not review the code for this resubmission; the code -and documentation- made available during the first submission was exemplary though.

Dear Reviewers,

Thank you very much for your thoughtful and constructive feedback. Your comments have substantially improved the clarity and rigor of our manuscript. A view-only version of the revised LaTeX document with track changes is available at the following link: <https://www.overleaf.com/read/wtfgbfkqmzn#b7f276>. We have also provided a PDF with highlighted revisions as a supplementary file.

We also apologize for the “file not found” message generated by the repository link (a known Anonymous github bug), but want to clarify that it does not prevent access to the data/code. The issue should now be resolved, but if the message reappears, all data and code remain fully accessible via the “Download repository” button in the top right. For convenience, we also include a .zip version as a supplementary file as a backup.

We have detailed our responses to each comment below. Key revisions include:

- 1) incorporating a new analysis evaluating how seasonal severity modulates the effect of group size on home range quality and fruit intake rate;
- 2) expanding our discussion to address the implications of these findings for interannual climate cycles;
- 3) clarifying methodological approaches, limitations, and interpretation of all response variables; and
- 4) validating model-derived metrics (encounter rate and revisitation rate) using empirical observations and independent measures in the supplementary material.

We greatly appreciate your contributions.

Reviewers' comments:

Reviewer #1 (Remarks to the Author):

This manuscript represents an excellent endeavor to leverage the power of a long-term dataset that included broad cross-sectional sampling, as well, to tease apart interrelated and confounding effects of within- and between-group competition. I'm unaware of any prior attempts to tackle similar analyses, and indeed I'm not certain there are many datasets available that could attempt to tackle this question with a similar time depth and cross-sectional breadth. I am, therefore, both very excited by the potential that the manuscript represents and inspired by the approach, but I have a few issues that I think may need to be addressed before the results presented fully track

with the authors' interpretation. Most critically, I think there is a bit of an overreach on the relative benefits of success in between-group contest competition and the costs of increased within-group contest and scramble, particularly as the authors then extrapolate to potential effects of an increasingly extreme climate within the site. I've noted below my concerns about the interpretations, all of which I think can be addressed through the inclusion of additional analyses and/or increased clarity of the text, if the case is that I am misinterpreting the results.

Major comments:

Lines 22-24: The first sentence suggests primacy of the benefits in between-group competition as a benefit of large group size, disregarding the impact of group size on predation risk. Overall, it feels that this potential benefit of group-living is given very little acknowledgement throughout the text. This may reflect the authors' perceptions of the relative importance of BGC vs. predation-sensitive behavioral strategies, but it does not feel fully representative of the broad literature on the topic.

Thank you for your comment. We agree that anti-predation benefits are an important factor associated with group size and should be acknowledged. We have therefore revised the opening sentence of this paragraph to better reflect the broader context:

Lines 23-25: "Ultimately, group size reflects trade-offs between the costs (e.g., within-group competition, disease transmission) and benefits (e.g., between-group competitive advantage, reduced predation risk) of group living."

With regards to our analyses, we would ideally include predation risk as a component, however, our dataset does not contain any convincing quantitative measure of predation pressure. Also, capuchins in this system have very few predators; only one predation event has been observed across the 33-year study.

Lines 112-117: For reference are there substantial increases that occurred in the absence of shifts in relative group size? Also, given the example in Figure 4 shows an 8-year step between the two samples, it appears that the "+1" component of $t+1$ is actually variable (might this be better called " $t+i$ "?). I understand that this may be necessary because of sampling distribution, but it may mean that the timing of the shift in relative group size and the timing of the encroachment are not as tightly correlated as suggested in the text.

Thank you very much for your feedback. We only found one out of 125 instances of substantial increases in proportional overlap (from <25% to >45%) where there was no change in relative group size. We completely agree with your second point, and have changed " $t+1$ " to " $t+i$ " in the Figure 4 caption, in equation 3, and on lines 136, 407 and 410.

Lines 142-144: I'm hesitant about saying that decreased home range overlap suggests territoriality

Line 164: changed from "suggesting increased territoriality" to "consistent with increased active defense".

Reference:

Burt, W. H. (1943). Territoriality and home range concepts as applied to mammals. *Journal of mammalogy*, 24(3), 346-352.

Lines 146-149: Again just because frequent encounters lead to spatial separation does not mean that the mechanism is territoriality or increased IGE costs for subordinate groups. Missing in this argument is evidence that the proportional shift during the dry season is greater for smaller groups, rather combining Lines 88-90 and lines 110-111 suggest that this is not the case; proportional home range overlap does not increase for small groups, and it's not just because they're contracting their ranges while large groups occupy the area. This finding can also be interpreted then as a decision by groups to avoid foraging in areas that are potentially already depleted. Frequent encounters could occur because groups are forced to revisit the area, but that they don't continue closer to the center of the range of the other because an encounter suggests that that BGS may be high. Can you repeat the annual analysis of directional range shift using the wet-dry season transition to get at this idea of whether large and small groups experience pressure from the neighbor unequally?

Thank you for this thoughtful comment. To clarify, we used "increased territorial defense" here in a broad sense: not strict territoriality with non-overlapping, patrolled boundaries, but rather an increase in behaviors consistent with active resource defense as resources become more spatially concentrated. As mentioned above, in the revised manuscript, we have changed the phrasing from "suggesting increased territoriality" to "consistent with increased active defense" on line 164.

Moreover, we agree that reduced spatial overlap in the dry season could, in principle, emerge from multiple non-exclusive mechanisms, including (a) active defense of high-value areas or (b) avoidance of recently depleted locations within overlap zones. However, in this system, we interpret the seasonal pattern of decreased overlap combined with increased encounters as more consistent with increased localized contest competition at key resource sites rather than with a depletion-avoidance mechanism, for the following reasons:

First, the depletion-avoidance argument does not provide a clear reasoning for why encounters should increase in the dry season. If groups reduce overlap because they are avoiding depleted areas, then one would expect that they should encounter one another less often. Instead, we observe the opposite pattern: encounters become more

frequent despite a reduction in overlap. This mismatch is difficult to reconcile with avoidance but is consistent with groups repeatedly returning to (and competing over) the same limited set of high-value dry-season resources (e.g., fruiting trees, water/shade at riparian sites).

Second, white-faced capuchins show strong xenophobia during intergroup encounters, which typically involve overt aggression (vocal threats, chases, and sometimes fatal attacks) (Gros-Louis et al. 2003; Perry 1996). Such escalated interactions are characteristic of direct (contest) competition, not responses to between-group indirect (scramble) competition. Thus, the rise in dry-season encounters is more consistent with increased active defense of spatially limited resources than with groups detecting depletion and withdrawing.

Moreover, we acknowledge that increased encounter rates do not suggest smaller/subordinate groups have increased IGE costs in the dry season. To clarify this, we revised the relevant section on lines 166 to 173, no longer suggesting that smaller groups have disproportionate costs during the dry season.

Lastly, regarding the suggestion to test whether large and small groups experience unequal neighbor pressure across the wet–dry transition, we appreciate this idea and attempted the proposed analysis. However, we lacked sufficient data to robustly compare enough dyads across consecutive wet and dry seasons. Even after relaxing the criteria for a substantial change in proportional overlap (change from <25% to >35% proportional overlap; original criteria was <25% to >45%) and for a clear difference in relative group size (absolute difference >3; original criteria was > 5), only three dyads met these conditions (using the original criteria only 2 dyads remained). In all three cases, the centroid of the larger group shifted toward the smaller group, consistent with larger groups displacing smaller ones during this transition.

That said, this approach appears limited for clarifying the mechanism in our case. The general reduction in overlap across dyads in the dry season, combined with higher encounter probabilities, indicates that between-group competition may become more localized at key sites within range boundaries rather than involving large-scale range displacement that would be detectable via centroid shifts. An asymmetric measure of overlap derived from the underlying utilization distributions could potentially be more informative, though, to our knowledge, no such method currently exists.

REFERENCES:

Gros-Louis, Julie, Susan Perry, and Joseph H. Manson. "Violent coalitionary attacks and intraspecific killing in wild white-faced capuchin monkeys (*Cebus capucinus*)." *Primates* 44.4 (2003): 341-346.

Perry, Susan. "Intergroup encounters in wild white-faced capuchins (*Cebus capucinus*).*" International Journal of Primatology* 17.3 (1996): 309-330.

Lines 198-201: This interpretation appears inconsistent with the data presented. Regardless of whether there is a difference between wet and dry season in the costs of larger groups size, it does appear to exist. Lower intake in large groups is not balanced by reduced travel costs, so one would predict lower average net energy gain in large groups. This does not therefore suggest that benefits from between-group contest competition outweigh the costs of within-group contest competition during the dry season, but that leveraging the greater competitive ability of the group means that they do not experience exacerbated costs of within-group contest competition during these seasons because they can gain access to the higher quality areas. Energetically they're still at a net negative relative to the smaller group.

Thank you for your feedback and we agree with your comment. The relevant paragraphs (lines 210-232 in revised manuscript) has been substantially revised in light of the new seasonal-severity analysis, and no longer suggests that larger groups are more likely to benefit disproportionately in the dry season.

Lines 238-240: Could you run an additional analysis including the severity of the dry season to get at this point?

Thank you very much for this helpful suggestion. We have implemented a new analysis addressing this idea. The results are described in section 2.2 (lines 98-114) and presented in Figure 2 b and c.

To provide sufficient background material on this new analysis, we have included new material in the revised Introduction on lines 56-59 and 76-78, as well as in section S1.3.2 of the Supplemental Information, pg. 6-9. The description for the methodology is provided on lines 317-326 and 443-448 in the Methods section. We interpret the new findings in the Discussion on lines 210-231.

The final sentences of the Discussion (lines 264–270) have also been revised from:

“Yet, increasingly severe droughts driven by climate change, coupled with growing habitat fragmentation, are rapidly altering tropical landscapes [92-95], making resources more clumped and monopolizable. Our findings reveal that such changes could raise the upper limit on group size by enhancing the benefits of between-group competition, thereby shifting the balance in favor of larger, more dominant groups in the future.”

To:

“Yet, increasingly intense and erratic ENSO cycles, together with ongoing habitat fragmentation, are rapidly altering tropical landscapes [80, 97-99], making resource availability increasingly unpredictable. Such climatic fluctuations appear to amplify within-group competition under some ecological contexts (e.g., El Niño dry seasons) while enhancing between-group advantages under others (e.g., La Niña dry seasons). These findings raise the question of whether ongoing climate change will destabilize these trade-offs, ultimately tilting the balance in favor of smaller or larger groups in the future.”

Minor comments:

Line 61: Should provide whether this is percent time or based on amount consumed.

We have clarified this in the revised text. The first reference (Carrera et al. 2025), which reports data from our study site (Lomas Barbudal), quantifies diet composition based on the percentage of time spent feeding (see their Figure S4). The second reference (McCabe & Fedigan, 2007), from a nearby long-term study at Santa Rosa, reports the proportion of total food items consumed (see their Figure 5). The revised sentence on lines 67-68 now reads:

“Capuchins are primarily frugivorous (~50-55%), with invertebrates comprising most of the remaining percentage (based both on time spent feeding [49] and proportion of food consumed [50]).”

Figure S3b: The key to the colors seems to be based on mean NDVI (same as represented in the y-axis), but the y-values where colors change are inconsistent. I think this means that I’m misreading the table, but I cannot come up with another value to base this on from the figure legend.

Thank you for pointing this out. I believe the discrepancy is because the color scale reflects the NDVI value of each point, while the connecting line inherits the color of the preceding point rather than varying continuously along its trajectory. We have clarified this in the figure caption for Figure S3b:

“Line colors correspond to the NDVI value at each observation (colors reflect the preceding point along each segment, not a continuous gradient along the line).”

Line 893-894: This assumption seems out of keeping with territoriality in some species wherein lower overlap is maintained specifically through frequent interactions at borders.

We appreciate and agree with your point, but our causal diagram was constructed *a priori* to reflect our initial hypotheses about the relationships among variables, rather than insights gained from our results or later literature review. At that stage, we

assumed that greater spatial overlap would provide more opportunities for between-group encounters, without much consideration of further mechanisms. We've therefore kept the original causal structure to maintain transparency in our pre-analysis reasoning, although we acknowledge that both our findings and some of the literature highlight a more nuanced relationship between overlap and encounter frequency than we originally anticipated.

To improve clarity, we added "a priori" in the sentence on line 928 in Supplementary Information.

Reviewer #1 (Remarks on code availability):

Attempts to review the data came up with "the requested file is not found".

Thank you for flagging this, and we apologize for the confusion. After looking into it, we found that this issue is due to a known bug in Anonymous GitHub that has affected several users (see: https://github.com/tdurieux/anonymous_github/issues/404). The message "The requested file is not found" appears only superficially and does not prevent the repository or its contents from being downloaded. This should now be resolved, but if the message reappears, please ignore it and click "Download repo" to access all files. For convenience, we have also provided a .zip version of the full repository as a backup.

Reviewer #2 (Remarks to the Author):

This study examines how environmental and demographic factors influence competitive interactions within and between primate social groups. Using a remarkable multi-year and multi-group dataset, the authors approach long-standing questions regarding the evolution of sociality with the latest in analytical techniques. The contributions of this work will be significant and I believe hold tremendous interest to the readership of this journal. Specific strengths include the exceptional clarity of writing, the well-coordinated analyses which add phenomenal robustness to their findings, and the uniqueness of their dataset. My critiques and questions are minimal. First, I wonder if the authors could succinctly address the distinction between within-group contest competition vs. within-group scramble competition. If the authors are perhaps choosing to focus solely on within-group scramble competition, it would be helpful to state this explicitly. Second, my understanding of Figure 1c raises the question of group id following a permanent fission (or fusion). It appears that, upon group fission, one of the fission products retained the former group's name. If this is correct, how did the authors determine which fission product retained the former

group's name and what are the implications of this group-level analyses? Lastly, and most minor, with regard to Table 1: I may be misunderstanding a fairly minor point, but how does proportional overlap provide insight into encroachment by neighbor as opposed to encroaching into a neighbor's space? My understanding is that these would be impossible to disentangle in the data available.

Thank you very much for the kind and constructive feedback. With regards to your first point, we have added the following on lines 9-10:

“which primarily reflects scramble (indirect) rather than contest (direct) competition among group members [5].”

Regarding your second point, this raises an important issue about how we operationalize groups in longitudinal field studies, which is something we've thought a lot about. In our study, when a permanent fission occurs, the group that retains the natal home range (invariably the larger faction) keeps the original ID of the parent group, whereas the group that splits off and forms a new home range is given a new identifier. After fission, the two groups function as distinct social units, so they are treated as independent within each season or year in our analyses. Over the full 33-year dataset, however, we acknowledge that daughter groups are not entirely independent lineages, as they share demographic and historical continuity with their parent groups. Our hierarchical modeling framework, which includes varying intercepts and slopes by group (and by dyad in the Social Relations Models), helps account for repeated measures within groups, though not for genealogical relatedness between them. Relatedness among groups could plausibly influence outcomes such as home range overlap or encounter rates, for instance if daughter groups remain spatially proximate to their natal range or, perhaps instead, avoid their parent group to reduce competition and conflict with kin. We've begun developing ways to model this more explicitly by treating group identity as a continuous variable using Gaussian process models, which could capture gradual divergence between related groups through a shared correlation structure. However, this remains out of the scope of the current study.

We have clarified how we treat group ID after fission events in the Methods on lines 275-277:

“When a permanent fission occurs, the group that retains the natal home range (invariably the larger, dominant faction) keeps the original identity of the parent group, whereas the group that splits off and forms a new home range is given a new identifier.”

To your third point, you are correct that proportional overlap (PO_m), while asymmetric (directional), does not exactly disentangle which group is encroaching upon the other, as both contribute to use of the shared space. We have revised the interpretation in Table 1 to “Extent of neighbour access to focal range.”

Reviewer #2 (Remarks on code availability):

Please see above.

Reviewer #3 (Remarks to the Author):

Overall assessment

This submission presents a beautiful case study of the Ecological Constraints Model of group size, showcasing how within- and between-group competition interact in shaping population-level patterns of space use in relation to group size and environmental seasonality. The dataset, spanning more than 3 decades of field-observations on 10+ neighbouring groups of a wild and free-ranging primate, is nothing short of spectacular, and the authors do it justice by the -overall- thoughtful application of state-of-the-art (Bayesian) linear models. The manuscript itself is clearly organized, has a compelling logical structure, and is very well-written. I further commend the authors for their exemplary supplementary materials that indeed appear to allow the reproduction of all statistical, and conceptual, analyses. In short, there is a lot here that warrants recommendation for publication in *Nature Ecology & Evolution*! I have taken the liberty below, however, to offer some critical suggestions to further strengthen this MS.

Minor concerns

I thought the use of the term “demography” as synonymous with group size throughout the manuscript was a little misleading. No consideration is given to other key demographic traits such as group structure in terms of relatedness, or age/sex composition, for example. If, for whatever justifiable reason, it was decided to restrict the analyses to group size only, that is totally fine, just don’t refer to it as demography then.

Thank you very much for your feedback. We agree that our use of the term demography could be clearer given that our analyses focus specifically on group size. We have revised the text accordingly, specifying “group size change” rather than “demographic change” on lines 50, 80, and 154. To clarify our broader conceptual framing, we now describe group size as “an emergent demographic trait” (lines 5–6), which provides context for our occasional use of demography in more general discussions later in the

manuscript. While “demography” encompasses a wider suite of traits (e.g., age–sex structure, relatedness, or reproductive rates), we use the term in its broader sense to include the population-level processes that shape group size (such as births, deaths, immigration, emigration, and fission events). Because our analyses focus on long-term patterns in group size as an emergent demographic property of social groups, we feel that retaining demography in certain contexts is appropriate.

The presentation of within- and between-group competition felt a little static on occasion, and important underlying social dynamics such as (the steepness of) within group dominance hierarchies, or coordination problems in mustering effective defence of a home range could (should?) be mentioned in the Introduction and/or discussion.

We greatly appreciate your feedback and agree that incorporating social dynamics and individual-level traits could add important nuance. However, these aspects are multifaceted (e.g., coordination in larger groups could either be hindered by collective action problems (Crofoot & Gilby, 2012) or enhanced through information pooling (King & Cowlshaw, 2007)) and deserving of a more thorough treatment than we could reasonably incorporate here without diluting the focus of the paper. We therefore consider this an important but separate avenue for future work. To acknowledge this limitation more explicitly, we revised the Discussion (lines 255–257) as follows:

Changed from: “Our framework provides the foundation for such extensions, and for a more nuanced incorporation of group composition (e.g., age–sex structure) and the resource landscape.”

To: “Our framework provides the foundation for such extensions, and for a more nuanced consideration of underlying social dynamics (e.g., age–sex structure, dominance hierarchies, collective action problems [96].”

References:

Crofoot, Margaret and Ian C. Gilby. "Cheating monkeys undermine group strength in enemy territory." *Proceedings of the National Academy of Sciences* 109.2 (2012): 501-505.

King AJ, Cowlshaw G. When to use social information: the advantage of large group size in individual decision making. *Biol Lett.* 2007 Apr 22;3(2):137-9.

The most serious ‘minor’ gripe I have with this study, however, is the idiosyncratic use of 89% HPDIs in the presentation of statistical results. I would rather see more conventional 50% and 95% HPDIs in graphs, for example, as well as the explicit reporting of the total proportion of the posterior probability in line with an effect being in a certain hypothesized direction (as you did in line 91). The authors now very much use 89% HPDIs as arbitrary cut-off values to decide whether they found an effect or not (quite contrary, I imagine, to what McElreath had in mind when introducing “yet another

arbitrary” criterion such as 89% CIs). It is further worth pointing out that the majority of reported effects would not have achieved “statistical significance” in the frequentist sense of a two-tailed hypothesis test at $\alpha = 0.05$. In fact, with the use of the 89% HPDI criterion in this study, the two-tailed frequentist equivalent of $\alpha = 0.22$ seems rather (overly?!) generous to make claims about having found an effect or not!

Thank you for your valuable feedback. As you correctly pointed out, our aim was to follow the convention by McElreath (Statistical Rethinking) who intentionally chose an unconventional level to remind readers that credible intervals should not be treated as significance thresholds. Our intention was therefore not to treat these intervals as decision cut-offs, but to communicate uncertainty and the shape of the posterior distribution. For this reason, we reported the proportion of the posterior greater than zero ($PP > 0$) in cases where interpretation might otherwise be ambiguous. In the revised manuscript, however, we report $PP > 0$ for all in-text summaries of effect sizes and have clarified our approach in the Methods (lines 481–486) as follows:

“We summarize parameter estimates using 89% Highest Posterior Density Intervals (HPDIs) and report the proportion of the posterior greater than zero ($PP > 0$), which represents the posterior probability that an effect is positive (values near 0 indicate strong support for a negative effect). Following McElreath [121], we use 89% intervals not as significance thresholds but as a convention to represent uncertainty while discouraging dichotomous interpretation of results (i.e., “significant” vs. “non-significant”).

Additionally, there are practical reasons for preferring the 89% level. Wider (95%) HPDIs can be more sensitive to Monte Carlo error and occasionally unstable across MCMC runs, as they depend more heavily on posterior tails (Kruschke, 2014). Using an 89% HPDI reduces this sensitivity without changing the overall conclusions, and the visual difference from 95% intervals is negligible. Finally, we may have misunderstood your last comment, but our understanding is that an 89% interval leaves 11% of the posterior mass in the tails (5.5 % in each), corresponding to a two-tailed $\alpha = 0.11$ in frequentist terms.

Reference:

Kruschke, J.K. (2014). *Doing Bayesian Data Analysis: A Tutorial with R, JAGS, and Stan* (2nd ed.). Academic Press.

Major concerns

My main reservation with this study, however, concerns the matter that at least 3 of the 7 response variables (daily path length, revisitation rate, and encounter rate) are highly derived, modelled estimates, rather than empirically observed values. This raises two concerns:

1. to varying degrees, I am unsure of the biological information content of these metrics, so this should be expanded upon

2. I can see at least 3 methodological caveats with specifying a modelled estimate as the response variable in a subsequent model. These potential risks need to be addressed more explicitly:

- Error propagation
- Lack of independence
- Double-counting of effects

As to the first point, I genuinely struggle to see, for example, the biological relevance of revisitation rate, calculated as it is here, as a global home range metric of “the mean revisitation rate per radial meter”. To me, revisitation rates make biological sense for depletable resources within a home range, not as a proxy of the rate of passing through a typical m² of home range. The interpretation of the calculated encounter rate is equally ambivalent: the ‘ctmm’ algorithm does not take into account that animals detect (and react to) each other’s proximity, well ahead of an encounter. As such, the estimated quantity represents an expectation under some sort of “null-model” of collision rates among inert (but not randomly, because autocorrelated, moving) particles. How this measure informs us about actual encounter rates of biological agents, or how it captures the extent of (experienced) between-group competition, is not clear.

To directly address the reviewer’s concern, we conducted two new empirical validation analyses (Supplementary Material, Section S1.8; pp. 29–32) comparing CTMM-derived quantities (encounter rates and revisitation rates) with independent, empirically derived measures (observed intergroup encounters and trajectory-based revisitations calculated with the Recurse R package). In both cases, CTMM-derived estimates showed strong alignment with their empirical counterparts. We also evaluated CTMM-derived daily path length against straight-line daily path length calculated at multiple sampling resolutions (5-, 10-, and 30-min), and all methods yielded the same relationship with group size (i.e., effect near zero). Because CTMM daily path length has already been rigorously validated (Noonan et al. 2019), we did not include this additional comparison in the manuscript or supplement but would be happy to do so if the editor or reviewers deem it helpful.

We agree that many of our response variables are derived from continuous-time movement models (home-range area, daily path length, encounter rate, revisitation rate). We chose this framework precisely because it accounts for irregular sampling schedules, variable sample sizes, and GPS location error (features inherent to our

dataset) and because it provides uncertainty estimates for all derived quantities. These properties make CTMM outputs directly comparable across groups, years, and seasons with differing sampling regimes. A substantial body of recent work has validated CTMM estimates of range area, daily path length, revisitation, and encounter rates across taxa (see references below), which gives us a robust methodological foundation to build on. In this context, we view CTMM as an appropriate and reliable tool for addressing our questions rather than a source of limitation.

Regarding biological interpretation, we now explicitly direct readers to Table 1 and Supplementary Section S1.4, where we describe the biological meaning and expected relationships for each metric. We have revised the main text (lines 75-76 and 80-81) to ensure that this information is easy to locate. Below we clarify the biological meaning of the variables that the reviewer raised concerns about and specify where the reader can find additional details within the manuscript:

Daily path length.

In the Introduction (lines 10-12), we clarify that travel distance is widely used as a proxy for daily energy expenditure (Carbone et al. 2005) and remains a central measure in tests of the Ecological Constraints Hypothesis (Wrangham et al. 1993).

Revisitation rate:

Supplementary Section S1.4 (lines 955-957) expands on how we interpret revisitation rate as related to area reuse and exploratory space use, and Section S1.7 explains why revisitation is relevant for testing ECM (see lines 1107-1114 of supplements). We also show that revisitation rate displays the same relationship with group size as “exploring new areas” measured via maximum displacement over longer timescales. This supports the interpretation, discussed in the main text (lines 204-209), that larger groups expand into underused areas across extended periods rather than increasing daily travel. In addition, we have revised the description of the CTMM-derived revisitation metric (lines 366-385) and direct readers to the new validation analysis demonstrating that this metric is monotonically related to trajectory-based revisit counts.

Encounter rate:

We appreciate the reviewer’s concern regarding the interpretation of CTMM encounter rates as potential “null” collision rates. The CTMM estimate is informed by each group’s empirically fitted movement process, which encompasses behavioral responses to other groups to the extent they shape movement trajectories. More importantly, our new validation analysis demonstrates strong agreement between CTMM-predicted encounter rates and observed encounters recorded by field researchers, suggesting that the model-based metric captures biologically realized intergroup interactions

rather than a purely theoretical expectation. In the revised manuscript, we revised our explanation of encounter rates in the Methods (lines 414-426) and direct readers to the supplementary validation analysis on line 427. Moreover, given potential non-random biases associated with the observational inter-group location dataset (highlighted in Supplementary Material, Section S1.8; lines 1178-1186), we consider CTMM encounter rates (which incorporate all available GPS data and yield uncertainty estimates) a more complete and comparable measure across groups and years.

References:

Fleming, Christen H., et al. "Overcoming the challenge of small effective sample sizes in home-range estimation." *Methods in Ecology and Evolution* 10.10 (2019): 1679-1689.

Noonan, Michael J., et al. "Scale-insensitive estimation of speed and distance traveled from animal tracking data." *Movement Ecology* 7.1 (2019): 35.

Fleming, C. H., et al. "Correcting for missing and irregular data in home-range estimation." *Ecological Applications* 28.4 (2018): 1003-1010.

Fleming, C. H., et al. "A comprehensive framework for handling location error in animal tracking data." *BioRxiv* (2020): 2020-06.

Noonan, Michael J., et al. "Estimating encounter location distributions from animal tracking data." *Methods in Ecology and Evolution* 12.7 (2021): 1158-1173.

Wrangham, Richard W., John L. Gittleman, and C. A. Chapman. "Constraints on group size in primates and carnivores: population density and day-range as assays of exploitation competition." *Behavioral ecology and Sociobiology* 32.3 (1993): 199-209.

Carbone C, Cowlshaw G, Isaac NJB, Rowcliffe JM. How Far Do Animals Go? Determinants of Day Range in Mammals. *Am Nat.* 2005;165(2):290–7.

As to the methodological caveats, I recognize that the authors add an error term around some (but not all?) of their modelled response variables when specifying them as the outcome in subsequent (Bayesian linear) models, presumably to alleviate concerns about error propagation? But, I'm not even sure to what extent modelled estimates can indeed be viewed as independent values, given that they are a function of the same underlying data and 'ctmm' model structure. I imagine that reporting on the independence of (Bayesian) model residuals would go a long way to alleviate this concern (on another but related note: wouldn't a multivariate approach, in which the correlation matrix amongst each of the outcome variables is explicitly estimated and incorporated, be preferable over fitting several models in isolation?). Lastly, the secondary (Bayesian linear) models may implicitly re-test patterns (e.g. the autocorrelation term estimated in the 'ctmm' algorithm, which is plausibly driven by group size, or environmental variation) that were explicitly built into the outcome by the

value generating process (the ctmm-model), thereby double-counting effects. This may result in biased and/or inflated estimates of effect sizes in the Bayesian models.

Thank you very much for your constructive feedback. Below we address each methodological concern independently:

Error propagation

We incorporate measurement-error modeling wherever feasible, allowing uncertainty from GPS location error and ctmm-derived confidence intervals to propagate to downstream inference. This is done for all ctmm-based response variables with validated and tractable uncertainty estimates (e.g., home range area, daily path length, encounter rates). Revisitation rate is the one exception, as formal uncertainty estimation for this metric is still under development. Other response variables (e.g., proportional overlap, fruit intake, mean home-range NDVI) are not direct ctmm outputs and therefore lack a coherent measurement-error structure. Adding an additional error layer in these cases would not be straightforward and could risk over-parameterizing already complex models (particularly the SRM).

Lack of independence

We appreciate the reviewer's concerns regarding potential non-independence across models. In our study, it is important to note that nearly all response variables are necessarily modeled using different subsets of data because they are derived from distinct ctmm models fitted to different data streams, each measured at its own temporal resolution (as detailed in the Methods). For example, daily path length is measured at the daily scale, whereas home range area and revisitation rate are measured at the annual or seasonal scale. Daily path length and revisitation rate rely on high-resolution GPS data available only from 2009–2020, while home range area and NDVI are estimated over a longer temporal range (1990–2023) using historical sleep-site locations and satellite imagery. Encounter rates are measured at the dyad-level, whereas the other ctmm-derived quantities are at the group-level. These differences mean that the response variables do not share common observational units, and therefore comparing model residuals or implementing a joint multivariate model across all response variables is neither practical nor conceptually appropriate.

Among the few response variables that do share observational units (home range area, home-range NDVI, and revisitation rate), a full multivariate model with correlated residuals requires that the responses share a common residual covariance structure, which can only be defined when all outcomes follow the same (typically Gaussian or Student-t) likelihood. In our case, the three outcomes arise from different non-Gaussian distributions (HRA: Gamma; NDVI: Beta; revisitation: lognormal), and therefore a single residual correlation matrix among them cannot be estimated using Bayesian R packages such as brms or rethinking. In principle, one could attempt to build such a

model directly in Stan, but doing so would require hand-coding a custom joint likelihood for Gamma-, Beta-, and lognormal-distributed outcomes, as well as restricting all analyses to the GPS years (2009–2020) so that the responses align. This would mean discarding the long-term data used for home range area and NDVI and fundamentally changing the structure of the analyses. For these reasons, a fully multivariate residual-covariance model was not feasible within the scope of this study.

More broadly, we do not expect strong dependence among most of our response variables. By design, they capture different dimensions of each group's movement ecology. These quantities are influenced by shared drivers such as group size and resource distribution, but they are not mathematically equivalent or different labels for the same underlying measure (similar to how trajectory-based movement metrics, such as path tortuosity and first-passage time, are related but not equivalent). For instance, groups can expand home range area without increasing daily path length, or alter revisitation patterns without changing encounter rates. Even in cases where two variables may be behaviorally related (e.g., home range area and revisitation rate) they capture different dimensions of how space is used: one describes the extent of the area a group uses, while the other describes how intensively that area is reused. In this sense, the response variables we use provide complementary rather than redundant information about how groups cope with resource competition and navigate their ecological and social environment.

Lastly, both revisitation and encounter rates are consistent with ctmm-independent measures (see Supplemental section S1.8), underscoring that these variables reflect genuine behavioral patterns rather than artifacts of a shared modeling framework.

Double-counting of effects

We appreciate the reviewer's concern, and we have tried to consider it carefully. However, we believe this issue does not apply to our analysis for several reasons. First, ctmm models do not incorporate group size or the other covariates used in our Bayesian regressions. The movement parameters estimated by ctmm (e.g., location and velocity autocorrelation timescales) arise solely from the geometry and temporal autocorrelation of the relocation paths; they are descriptive summaries of the animals' movement, not model-based estimates conditioned on group size or environmental features. Thus, the downstream regressions do not "re-test" effects that were built into the ctmm estimation process.

Second, our analytical workflow follows a standard two-stage approach widely used in ecology: first estimating biologically meaningful quantities from raw data, then modeling ecological predictors of variation in those estimates. This is directly analogous to deriving home-range UDs before fitting resource-selection functions, or estimating survival from capture–recapture models before relating it to climate, etc.

Because the first-stage ctmm models and the second-stage Bayesian models use distinct likelihoods applied to different observational units, there is no reuse of predictors or duplication of statistical information.

Third, we explicitly propagate measurement error from ctmm estimates (where validated uncertainty estimates exist), which makes the regression estimates more conservative, not inflated. This approach avoids treating ctmm-derived quantities as if they were error-free and reduces the risk of overstated effect sizes.

Finally, even if group size or environmental conditions influence the autocorrelation structure of movement, this reflects genuine biological variation rather than a statistical artifact. The autocorrelation timescales estimated by ctmm describe the animals' movement behaviour. Thus, if group size does indeed affect these parameters, then this is precisely the relationship our regression models are intended to evaluate; it is not something introduced or enforced by the ctmm fitting process.

Recommendation

I hope that by addressing the concerns expressed above, the authors are able to produce a revised version of their manuscript, that I would -wholeheartedly- be willing to recommend for publication!

Line by line comments

Line 23 -24

“mediated by additional ecological and social factors” Or, alternatively, modulated by local heterogeneity.

Thank you for this suggestion. We retain the general wording here but address local heterogeneity in the following sentences.

Lines 28-29 changed from:

“Simultaneously, environmental conditions and the spatiotemporal distribution of resources influence ...”

To:

“Simultaneously, local heterogeneity and shifting environmental conditions influence ..”

Figure 2 (an all similar figures of effect sizes)

I really like that you show the posterior distributions in grey, but would love to see medians along with 50 and 95% HPDIs, rather than -the equally arbitrary, but far less common- 89% HPDIs. You now very much seem to use these 89% values as “critical values”, which is not in line with your Bayesian approach.

This point is addressed in our response to the reviewer's "minor concerns" above, and on lines 481-486 of the revised manuscript.

Line 91

Here you actually report the proportion of the posterior probability that supports your hypothesis ($PP > 0 = 0.94$). I would like to see this reported for all effects!

$PP > 0$ is now reported for all in-text summaries of effect sizes in the revised manuscript.

Line 123

How do these predicted encounter rates relate to observed encounter rates: was there any way to empirically validate these derived estimates?

In the revised supplementary material (section S1.8; pg. 29-32), we provide an empirical validation of ctmm-derived encounter rates.

Line 133

Group demography is not the same as group size!

Line 154 of revised manuscript: Changed from "group demography" to "group size".

Line 174 -176

But you didn't find evidence in your study that home ranges of smaller groups are marginal in terms of lower NDVI values.

We did not intend "marginal" to imply lower-quality habitat, so we have removed the term. The revised text (lines 197-199) changed from:

"Such conditions allow dominant groups to access new resources with minimal conflict while enabling smaller groups to persist in marginal or underused zones between the ranges of their more powerful neighbors."

To:

"Such conditions allow dominant groups to access new resources with minimal conflict while enabling smaller groups to persist in underused zones between the ranges of their more powerful neighbors."

Line 185 – 187

Is this bit, alluding to advanced cognitive abilities, required? You certainly can't say anything about this on the basis of the data you present in this study, so it reads a little overly speculative.

We agree and have removed this sentence in the revised manuscript. Thank you.

Line 234

“these constraints are not fixed” I’m not aware of anyone ever claiming they were?

Lines 260-261 of revised manuscript changed from “these constraints are not fixed” to “the strength of these constraints is context-dependent.”

Line 238 -240

Is the link to climate change required here? It's rather speculative to argue that, on the basis of your study, habitat fragmentation should favour larger groups. There are many other reasons why this may or may not be the case.

Thank you for this comment. We retain the link to climate change because we feel it is relevant to our findings, particularly in light of the new seasonal-severity analysis added in response to Reviewer 1’s comments. Nonetheless, we have revised the concluding paragraph (lines 262-268) to frame this point as an open question rather than suggesting that larger groups are likely to be disproportionately favored.

Line 255 – 256

Nice that you could validate the soundness of this approach: would have been great to see similar validations of your other response variables.

Please see section S1.8 in the revised supplementary information (pg. 29-32) for a new validation analysis of encounter and revisitation rates.

Line 333

What does a single value for an entire home range mean... Are you sure that a similar global value cannot come about through very different local scenario’s within a home range?!

Thank you for this comment. We have clarified in the Methods (lines 366-385) that the revisitation estimate is first calculated at every location on the UD grid within a group’s range, capturing local differences in how often different areas are reused. The single value we report is the average of these local estimates and describes the group’s typical return rate to previously used locations (that is, how often the group comes back to areas it has already visited versus moving into new areas). To further address this point, we have also added a comparison with a trajectory-based revisitation measure in the Supplement (section S1.8; pg. 29-32).

Line 348

How did you handle triadic home range overlap? It seems possible, in theory, for a group to have its different dyadic POs to sum up to over 100% with your current definition. Is this problematic?

Thank you for this comment. To clarify, a focal group's overlaps with different neighbors are not expected to sum to 100%; they can exceed 1 without causing any analytical issues. Our analysis independently examines the relationship between a focal group and each of its neighbors, as well as how each dyadic overlap changes over time. The social relations model is well suited to this because it captures these dyadic relationships while accounting for repeated measures and the fact that groups share neighbors. This approach is preferable to methods that compute "area exclusivity" by subtracting the overlap of all neighbors from the focal group's range, as such methods assume that all groups in a population are monitored, which is rarely feasible. As noted in the Discussion (lines 252-255), extending this framework to include the combined influence of multiple neighbors (including unobserved groups) is an important direction for future work, but to the best of our knowledge current methods do not yet provide a general solution to this challenge.

Line 360

How many of these dyads were there?

In the revised manuscript on lines 408-409, we now specify that there are 63 cross-year comparisons across 11 unique group pairs that fit the criteria for the range-shift analysis.

Line 372

This definition assumes that groups are impervious to each other's proximity or general whereabouts, a very unrealistic assumption.

We have revised the subsection in the methods describing the quantification and definition of encounter rates (lines 414-427), clarifying that this metric is not a count of empirical instances in which groups simultaneously enter a defined encounter radius, but rather a model-based probability estimate that is not sensitive to the chosen radius. We also show that these estimates accurately reflect field-observed encounter frequencies in supplemental section S1.8.

Line 373

"These values are relative and only interpretable when compared across dyads". But, are they even interpretable in a biologically meaningful way at all?

This point is addressed in our response to the reviewer's first major concern about biological meaning/interpretability of response variables. We have revised the description of encounter rates to improve interpretability (lines 414-427) and show that these estimates accurately reflect field-observed encounter frequencies in supplemental section S1.8.

Line 377

I wonder whether the dependency between the different outcomes could/should have been taken into account, in an explicitly multivariate model framework for example.

This point is addressed in our response to the reviewer's second major concern about lack of independence.

Reviewer #3 (Remarks on code availability):

(see comments under "overall assessment")

We thank all reviewers for their thorough and generous assessment of the revised manuscript. As the reviewers raised no outstanding concerns, we have no further responses to provide.

Reviewer #2:

Remarks to the Author:

In my second review of this manuscript, I find the authors have enhanced their study not only with relatively minor text updates but - more significantly - with thoughtfully executed additional analyses focused on severe seasonal variability and model-derived metric validation. These revisions and additional analyses add much to an already strong manuscript. Specifically, several particular yet significant elements of the manuscript (e.g. the concept of territoriality vs. range overlap) have been clarified and the new analyses wonderfully extend the results presented in the first submission. In my opinion, the authors have successfully addressed the reviewer concerns either through manuscript edits or, in some cases, detailed clarifications in their response letter. As noted in my prior review, I believe this study has been superbly designed and executed and that its findings will be of great interest to the journal's readership.

Remarks on code availability:

I continue to have trouble with the Github link, but have reviewed the contents of the posted zip file to the best of my ability.

Reviewer #3:

Remarks to the Author:

Congratulations to the authors on doing an excellent job addressing the concerns raised about the previous version of their manuscript. I particularly appreciate the addition of new analyses to empirically validate the cttm-derived movement metrics: this must have been a lot of work, but I believe really strengthens their study. Although I remain somewhat cautious about the interpretability of a "globally averaged" revisitation rate, I very enthusiastically recommend this paper for publication at this point: a beautiful, and well-executed study!

Remarks on code availability:

I did not review the code for this resubmission; the code -and documentation- made available during the first submission was exemplary though.